# Dual Decomposed Learning with Factorwise Oracles for Structural SVMs of Large Output Domain

**Ian E.H. Yen** [†]   **Xiangru Huang** [‡]   **Kai Zhong** [‡]   **Ruohan Zhang** [‡]
**Pradeep Ravikumar** [†]   **Inderjit S. Dhillon** [‡]
[†] Carnegie Mellon University   [‡] University of Texas at Austin

## Abstract

Many applications of machine learning involve structured outputs with large domains, where learning of a structured predictor is prohibitive due to repetitive calls to an expensive inference oracle. In this work, we show that by decomposing training of a Structural Support Vector Machine (SVM) into a series of multiclass SVM problems connected through messages, one can replace an expensive structured oracle with Factorwise Maximization Oracles (FMOs) that allow efficient implementation of complexity sublinear to the factor domain. A Greedy Direction Method of Multiplier (GDMM) algorithm is then proposed to exploit the sparsity of messages while guarantees convergence to $\epsilon$ sub-optimality after $O(\log(1/\epsilon))$ passes of FMOs over every factor. We conduct experiments on chain-structured and fully-connected problems of large output domains, where the proposed approach is orders-of-magnitude faster than current state-of-the-art algorithms for training Structural SVMs.

## 1 Introduction

Structured prediction has become prevalent with wide applications in Natural Language Processing (NLP), Computer Vision, and Bioinformatics to name a few, where one is interested in outputs of strong interdependence. Although many dependency structures yield intractable inference problems, approximation techniques such as convex relaxations with theoretical guarantees [10, 14, 7] have been developed. However, solving the relaxed problems (LP, QP, SDP, etc.) is computationally expensive for factor graphs of large output domain and results in prohibitive training time when embedded into a learning algorithm relying on inference oracles [9, 6]. For instance, many applications in NLP such as Machine Translation [3], Speech Recognition [21], and Semantic Parsing [1] have output domains as large as the size of vocabulary, for which the prediction of even a single sentence takes considerable time.

One approach to avoid inference during training is by introducing a loss function conditioned on the given labels of neighboring output variables [15]. However, it also introduces more variance to the estimation of model and could degrade testing performance significantly. Another thread of research aims to formulate parameter learning and output inference as a joint optimization problem that avoids treating inference as a subroutine [12, 11]. In this appraoch, the structured hinge loss is reformulated via dual decomposition, so both messages between factors and model parameters are treated as first-class variables. The new formulation, however, does not yield computational advantage due to the constraints entangling the two types of variables. In particular, [11] employs a hybrid method (DLPW) that alternatively optimizes model parameters and messages, but the algorithm is not significantly faster than directly performing stochastic gradient on the structured hinge loss. More recently, [12] proposes an approximate objective for structural SVMs that leads to an algorithm considerably faster than DLPW on problems requiring expensive inference. However, the

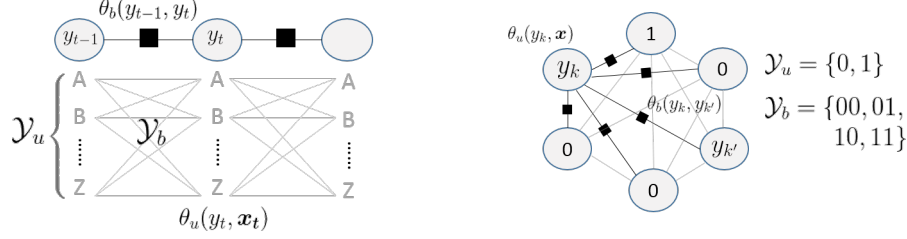

Figure 1: (left) Factors with large output domains in Sequence Labeling. (right) Large number of factors in a Correlated Multilabel Prediction problem. Circles denote variables and black boxes denote factors. ($\mathcal{Y}_u$: domain of unigram factor. $\mathcal{Y}_b$: domain of bigram factor.)

approximate objective requires a trade-off between efficiency and approximation quality, yielding an $O(1/\epsilon^2)$ overall iteration complexity for achieving $\epsilon$ sub-optimality.

The contribution of this work is twofold. First, we propose a Greedy Direction Method of Multiplier (GDMM) algorithm that decomposes the training of a structural SVM into factorwise multiclass SVMs connected through sparse messages confined to the active labels. The algorithm guarantees an $O(\log(1/\epsilon))$ iteration complexity for achieving an $\epsilon$ sub-optimality and each iteration requires only one pass of *Factorwise Maximization Oracles (FMOs)* over every factor. Second, we show that the FMO can be realized in time sublinear to the cardinality of factor domains, hence is considerably more efficient than a structured maximization oracle when it comes to large output domain. For problems consisting of numerous binary variables, we further give realization of a joint FMO that has complexity sublinear to the number of factors. We conduct experiments on both chain-structured problems that allow exact inference and fully-connected problems that rely on Linear Program relaxations, where we show the proposed approach is orders-of-magnitude faster than current state-of-the-art training algorithms for Structured SVMs.

## 2   Problem Formulation

Structured prediction aims to predict a set of outputs $\boldsymbol{y} \in \mathcal{Y}(\boldsymbol{x})$ from their interdependency and inputs $\boldsymbol{x} \in \mathcal{X}$. Given a feature map $\boldsymbol{\phi}(\boldsymbol{x}, \boldsymbol{y}) : \mathcal{X} \times \mathcal{Y}(\boldsymbol{x}) \to \mathbb{R}^d$ that extracts relevant information from $(\boldsymbol{x}, \boldsymbol{y})$, a linear classifier with parameters $\boldsymbol{w}$ can be defined as $h(\boldsymbol{x}; \boldsymbol{w}) = arg\max_{\boldsymbol{y} \in \mathcal{Y}(\boldsymbol{x})} \langle \boldsymbol{w}, \boldsymbol{\phi}(\boldsymbol{x}, \boldsymbol{y}) \rangle$, where we estimate the parameters $\boldsymbol{w}$ from a training set $\mathcal{D} = \{(\boldsymbol{x}_i, \bar{\boldsymbol{y}}_i)\}_{i=1}^n$ by solving a regularized *Empirical Risk Minimization (ERM)* problem

$$\min_{\boldsymbol{w}} \quad \frac{1}{2}\|\boldsymbol{w}\|^2 + C\sum_{i=1}^n L(\boldsymbol{w}; \boldsymbol{x}_i, \bar{\boldsymbol{y}}_i) . \tag{1}$$

In case of a Structural SVM [19, 20], we consider the structured hinge loss

$$L(\boldsymbol{w}; \boldsymbol{x}, \bar{\boldsymbol{y}}) = \max_{\boldsymbol{y} \in \mathcal{Y}(\boldsymbol{x})} \langle \boldsymbol{w}, \ \boldsymbol{\phi}(\boldsymbol{x}, \boldsymbol{y}) - \boldsymbol{\phi}(\boldsymbol{x}, \bar{\boldsymbol{y}}) \rangle + \delta(\boldsymbol{y}, \bar{\boldsymbol{y}}), \tag{2}$$

where $\delta(\boldsymbol{y}, \bar{\boldsymbol{y}}_i)$ is a task-dependent error function, for which the Hamming distance $\delta_H(\boldsymbol{y}, \bar{\boldsymbol{y}}_i)$ is commonly used. Since the size of domain $|\mathcal{Y}(\boldsymbol{x})|$ typically grows exponentially with the number of output variables, the tractability of problem (1) lies in the decomposition of the responses $\langle \boldsymbol{w}, \boldsymbol{\phi}(\boldsymbol{x}, \boldsymbol{y}) \rangle$ into several factors, each involving only a few outputs. The factor decomposition can be represented as a bipartite graph $G(\mathcal{F}, \mathcal{V}, \mathcal{E})$ between factors $\mathcal{F}$ and variables $\mathcal{V}$, where an edge $(f, j) \in \mathcal{E}$ exists if the factor $f$ involves the variable $j$. Typically, a set of factor templates $\mathcal{T}$ exists so that factors of the same template $F \in \mathcal{T}$ share the same feature map $\boldsymbol{\phi}_F(.)$ and parameter vector $\boldsymbol{w}_F$. Then the response on input-output pair $(\boldsymbol{x}, \boldsymbol{y})$ is given by

$$\langle \boldsymbol{w}, \ \boldsymbol{\phi}(\boldsymbol{x}, \boldsymbol{y}) \rangle = \sum_{F \in \mathcal{T}} \sum_{f \in F(\boldsymbol{x})} \langle \boldsymbol{w}_F, \boldsymbol{\phi}_F(\boldsymbol{x}_f, \boldsymbol{y}_f) \rangle, \tag{3}$$

where $F(\boldsymbol{x})$ denotes the set of factors on $\boldsymbol{x}$ that share a template $F$, and $\boldsymbol{y}_f$ denotes output variables relevant to factor $f$ of domain $\mathcal{Y}_f = \mathcal{Y}_F$. We will use $\mathcal{F}(\boldsymbol{x})$ to denote the union of factors of different templates $\{F(\boldsymbol{x})\}_{F \in \mathcal{T}}$. Figure 1 shows two examples that both have two factor templates

(i.e. unigram and bigram) for which the responses have decomposition $\sum_{f\in u(\boldsymbol{x})}\langle \boldsymbol{w}_u, \phi_u(\boldsymbol{x}_f, y_f)\rangle + \sum_{f\in b(\boldsymbol{x})}\langle \boldsymbol{w}_b, \phi_b(y_f)\rangle$. Unfortunately, even with such decomposition, the maximization in (2) is still computationally expensive. First, most of graph structures do not allow exact maximization, so in practice one would minimize an upper bound of the original loss (2) obtained from relaxation [10, 18]. Second, even for the relaxed loss or a tree-structured graph that allows polynomial-time maximization, its complexity is at least linear to the cardinality of factor domain $|\mathcal{Y}_f|$ times the number of factors $|\mathcal{F}|$. This results in a prohibitive computational cost for problems with large output domain. As in Figure 1, one example has a factor domain $|\mathcal{Y}_b|$ which grows quadratically with the size of output domain; the other has the number of factors $|\mathcal{F}|$ which grows quadratically with the number of outputs. A key observation of this paper is, in contrast to the structural maximization (2) that requires larger extent of exploration on locally suboptimal assignments in order to achieve global optimality, the *Factorwise Maximization Oracle (FMO)*

$$\boldsymbol{y}_f^* := \underset{\boldsymbol{y}_f}{argmax}\ \langle \boldsymbol{w}_F, \phi(\boldsymbol{x}_f, \boldsymbol{y}_f)\rangle \tag{4}$$

can be realized in a more efficient way by maintaining data structures on the factor parameters $\boldsymbol{w}_F$. In the next section, we develop globally-convergent algorithms that rely only on FMO, and provide realizations of *message-augmented FMO* with cost sublinear to the size of factor domain or to the number of factors.

## 3  Dual-Decomposed Learning

We consider an upper bound of the loss (2) based on a Linear Program (LP) relaxation that is tight in case of a tree-structured graph and leads to a tractable approximation for general factor graphs [11, 18]:

$$L^{LP}(\boldsymbol{w}; \boldsymbol{x}, \bar{\boldsymbol{y}}) = \max_{(\boldsymbol{q},\boldsymbol{p})\in\mathcal{M}_L} \sum_{f\in\mathcal{F}(\boldsymbol{x})} \langle \boldsymbol{\theta}_f(\boldsymbol{w}), \boldsymbol{q}_f\rangle \tag{5}$$

where $\boldsymbol{\theta}_f(\boldsymbol{w}) := \big(\langle \boldsymbol{w}_F, \phi_F(\boldsymbol{x}_f, \boldsymbol{y}_f) - \phi_F(\boldsymbol{x}_f, \bar{\boldsymbol{y}}_f)\rangle + \delta_f(\boldsymbol{y}_f, \bar{\boldsymbol{y}}_f)\big)_{\boldsymbol{y}_f\in\mathcal{Y}_f}$. $\mathcal{M}_L$ is a polytope that constrains $\boldsymbol{q}_f$ in a $|\mathcal{Y}_f|$-dimensional simplex $\Delta^{|\mathcal{Y}_f|}$ and also enforces local consistency:

$$\mathcal{M}_L := \left\{ \begin{array}{l} \boldsymbol{q} = (\boldsymbol{q}_f)_{f\in\mathcal{F}(\boldsymbol{x})} \\ \boldsymbol{p} = (\boldsymbol{p}_j)_{j\in\mathcal{V}(\boldsymbol{x})} \end{array} \middle| \begin{array}{ll} \boldsymbol{q}_f \in \Delta^{|\mathcal{Y}_f|}, & \forall f\in F(\boldsymbol{x}), \forall F\in\mathcal{T} \\ M_{jf}\boldsymbol{q}_f = \boldsymbol{p}_j, & \forall (j,f)\in\mathcal{E}(\boldsymbol{x}) \end{array} \right\},$$

where $M_{jf}$ is a $|\mathcal{Y}_j|$ by $|\mathcal{Y}_f|$ matrix that has $M_{jf}(y_j, \boldsymbol{y}_f) = 1$ if $y_j$ is consistent with $\boldsymbol{y}_f$ (i.e. $y_j = [\boldsymbol{y}_f]_j$) and $M_{jf}(y_j, \boldsymbol{y}_f) = 0$ otherwise. For a tree-structured graph $G(\mathcal{F}, \mathcal{V}, \mathcal{E})$, the LP relaxation is tight and thus loss (5) is equivalent to (2). For a general factor graph, (5) is an upper bound on the original loss (2). It is observed that parameters $\boldsymbol{w}$ learned from the upper bound (5) tend to tightening the LP relaxation and thus in practice lead to tight LP in the testing phase [10]. Instead of solving LP (5) as a subroutine, a recent attempt formulates (1) as a problem that optimizes $(\boldsymbol{p}, \boldsymbol{q})$ and $\boldsymbol{w}$ jointly via dual decomposition [11, 12]. We denote $\boldsymbol{\lambda}_{jf}$ as dual variables associated with constraint $M_{jf}\boldsymbol{q}_f = \boldsymbol{p}_j$, and $\boldsymbol{\lambda}_f := (\boldsymbol{\lambda}_{jf})_{j\in\mathcal{N}(f)}$ where $\mathcal{N}(f) = \{j \mid (j,f)\in\mathcal{E}\}$. We have

$$L^{LP}(\boldsymbol{w}; \boldsymbol{x}, \bar{\boldsymbol{y}}) = \max_{\boldsymbol{q},\boldsymbol{p}} \min_{\boldsymbol{\lambda}} \sum_{f\in\mathcal{F}(\boldsymbol{x})} \langle \boldsymbol{\theta}_f(\boldsymbol{w}), \boldsymbol{q}_f\rangle + \sum_{j\in\mathcal{N}(f)} \langle \boldsymbol{\lambda}_{jf}, M_{jf}\boldsymbol{q}_f - \boldsymbol{p}_j\rangle \tag{6}$$

$$= \min_{\boldsymbol{\lambda}\in\Lambda} \sum_{f\in\mathcal{F}(\boldsymbol{x})} \max_{\boldsymbol{q}_f\in\Delta^{|\mathcal{Y}_f|}} \Big(\boldsymbol{\theta}_f(\boldsymbol{w}) + \sum_{j\in\mathcal{N}(f)} M_{jf}^T\boldsymbol{\lambda}_{jf}\Big)^T\boldsymbol{q}_f \tag{7}$$

$$= \min_{\boldsymbol{\lambda}\in\Lambda} \sum_{f\in\mathcal{F}(\boldsymbol{x})} \left( \max_{\boldsymbol{y}_f\in\mathcal{Y}_f} \theta_f(\boldsymbol{y}_f; \boldsymbol{w}) + \sum_{j\in\mathcal{N}(f)} \lambda_{jf}([\boldsymbol{y}_f]_j) \right) = \min_{\boldsymbol{\lambda}\in\Lambda} \sum_{f\in\mathcal{F}(\boldsymbol{x})} L_f(\boldsymbol{w}; \boldsymbol{x}_f, \bar{\boldsymbol{y}}_f, \boldsymbol{\lambda}_f) \tag{8}$$

where (7) follows the strong duality, and the domain $\Lambda = \left\{ \boldsymbol{\lambda} \middle| \sum_{(j,f)\in\mathcal{E}(\boldsymbol{x})} \boldsymbol{\lambda}_{jf} = \boldsymbol{0}, \forall j\in\mathcal{V}(\boldsymbol{x}) \right\}$ follows the maximization w.r.t. $\boldsymbol{p}$ in (6). The result (8) is a loss function $L_f(.)$ that penalizes the response of each factor separately given $\boldsymbol{\lambda}_f$. The ERM problem (1) can then be expressed as

$$\min_{\boldsymbol{w},\boldsymbol{\lambda}\in\Lambda} \sum_{F\in\mathcal{T}} \left( \frac{1}{2}\|\boldsymbol{w}_F\|^2 + C\sum_{f\in F} L_f(\boldsymbol{w}_F; \boldsymbol{x}_f, \bar{\boldsymbol{y}}_f, \boldsymbol{\lambda}_f) \right), \tag{9}$$

---

**Algorithm 1** Greedy Direction Method of Multiplier

---
0. Initialize $t = 0$, $\boldsymbol{\alpha}^0 = \mathbf{0}$, $\boldsymbol{\lambda}^0 = \mathbf{0}$ and $\mathcal{A}^0 = \mathcal{A}^{init}$.
   **for** $t = 0, 1, \ldots$ **do**
      1. Compute $(\boldsymbol{\alpha}^{t+1}, \mathcal{A}^{t+1})$ via one pass of Algorithm 2, 3, or 4.
      2. $\boldsymbol{\lambda}_{jf}^{t+1} = \boldsymbol{\lambda}_{jf}^t + \eta \left( M_{jf} \boldsymbol{\alpha}_f^{t+1} - \boldsymbol{\alpha}_j^{t+1} \right)$, $j \in \mathcal{N}(f)$, $\forall f \in \mathcal{F}$.
   **end for**

---

where $F = \bigcup_{i=1}^N F(\boldsymbol{x}_i)$ and $\mathcal{F} = \bigcup_{F \in \mathcal{T}} F$. The formulation (9) has an insightful interpretation: each factor template $F$ learns a multiclass SVM given by parameters $\boldsymbol{w}_F$ from factors $f \in F$, while each factor is augmented with messages $\boldsymbol{\lambda}_f$ passed from all variables related to $f$.

Despite the insightful interpretation, formulation (9) does not yield computational advantage directly. In particular, the non-smooth loss $L_f(.)$ entangles parameters $\boldsymbol{w}$ and messages $\boldsymbol{\lambda}$, which leads to a difficult optimization problem. Previous attempts to solve (9) either have slow convergence [11] or rely on an approximation objective [12]. In the next section, we propose a *Greedy Direction Method of Multiplier (GDMM)* algorithm for solving (9), which achieves $\epsilon$ sub-optimality in $O(\log(1/\epsilon))$ iterations while requiring only one pass of FMOs for each iteration.

### 3.1 Greedy Direction Method of Multiplier

Let $\alpha_f(\boldsymbol{y}_f)$ be dual variables for the factor responses $z_f(\boldsymbol{y}_f) = \langle \boldsymbol{w}, \boldsymbol{\phi}(\boldsymbol{x}_f, \boldsymbol{y}_f) \rangle$ and $\{\boldsymbol{\alpha}_j\}_{j \in \mathcal{V}}$ be that for constraints in $\Lambda$. The dual problem of (9) can be expressed as [1]

$$
\begin{aligned}
\min_{\boldsymbol{\alpha}_f \in \Delta^{|\mathcal{Y}_f|}} \quad & G(\boldsymbol{\alpha}) := \frac{1}{2} \sum_{F \in \mathcal{T}} \|\boldsymbol{w}_F(\boldsymbol{\alpha})\|^2 - \sum_{j \in \mathcal{V}} \boldsymbol{\delta}_j^T \boldsymbol{\alpha}_j \\
s.t. \quad & M_{jf} \boldsymbol{\alpha}_f = \boldsymbol{\alpha}_j, \ j \in \mathcal{N}(f), f \in \mathcal{F}. \\
& \boldsymbol{w}_F(\boldsymbol{\alpha}) = \sum_{f \in F} \Phi_f^T \boldsymbol{\alpha}_f
\end{aligned}
\tag{10}
$$

where $\boldsymbol{\alpha}_f$ lie in the shifted simplex

$$
\Delta^{|\mathcal{Y}_f|} := \left\{ \boldsymbol{\alpha}_f \mid \alpha_f(\bar{\boldsymbol{y}}_f) \leq C, \ \alpha_f(\boldsymbol{y}_f) \leq 0, \ \forall \boldsymbol{y}_f \neq \bar{\boldsymbol{y}}_f, \ \sum_{\boldsymbol{y}_f \in \mathcal{Y}_f} \alpha_f(\boldsymbol{y}_f) = 0. \right\}.
\tag{11}
$$

Problem (10) can be interpreted as a summation of the dual objectives of $|\mathcal{T}|$ multiclass SVMs (each per factor template), connected with consistency constraints. To minimize (10) one factor at a time, we adopt a *Greedy Direction Method of Multiplier (GDMM)* algorithm that alternates between minimizing the *Augmented Lagrangian* function

$$
\min_{\boldsymbol{\alpha}_f \in \Delta^{|\mathcal{Y}_f|}} \mathcal{L}(\boldsymbol{\alpha}, \boldsymbol{\lambda}^t) := G(\boldsymbol{\alpha}) + \frac{\rho}{2} \sum_{j \in \mathcal{N}(f), f \in \mathcal{F}} \left\| \boldsymbol{m}_{jf}(\boldsymbol{\alpha}, \boldsymbol{\lambda}^t) \right\|^2 - \|\boldsymbol{\lambda}_{jf}^t\|^2
\tag{12}
$$

and updating the Lagrangian Multipliers (of consistency constraints)

$$
\boldsymbol{\lambda}_{jf}^{t+1} = \boldsymbol{\lambda}_{jf}^t + \eta \left( M_{jf} \boldsymbol{\alpha}_f - \boldsymbol{\alpha}_j \right). \ \forall j \in \mathcal{N}(f), \ f \in \mathcal{F},
\tag{13}
$$

where $\boldsymbol{m}_{jf}(\boldsymbol{\alpha}, \boldsymbol{\lambda}^t) = M_{jf} \boldsymbol{\alpha}_f - \boldsymbol{\alpha}_j + \boldsymbol{\lambda}_{jf}^t$ plays the role of messages between $|\mathcal{T}|$ multiclass problems, and $\eta$ is a constant step size. The procedure is outlined in Algorithm 1. The minimization (12) is conducted in an approximate and greedy fashion, in the aim of involving as few dual variables as possible. We discuss two greedy algorithms that suit two different cases in the following.

**Factor of Large Domain** For problems with large factor domains, we minimize (12) via a variant of *Frank-Wolfe* algorithm with *away steps* (AFW) [8], outlined in Algorithm 2. The AFW algorithm maintains the iterate $\boldsymbol{\alpha}^t$ as a linear combination of bases constructed during iterates

$$
\boldsymbol{\alpha}^t = \sum_{\boldsymbol{v} \in \mathcal{A}^t} c_{\boldsymbol{v}}^t \boldsymbol{v}, \quad \mathcal{A}^t := \{\boldsymbol{v} \mid c_{\boldsymbol{v}}^t \neq 0\}
\tag{14}
$$

| **Algorithm 2** Away-step Frank-Wolfe (AFW) | **Algorithm 3** Block-Greedy Coordinate Descent |
|---|---|
| **repeat**<br>    1. Find a greedy direction $\boldsymbol{v}^+$ satisfying (15).<br>    2. Find an away direction $\boldsymbol{v}^-$ satisfying (16).<br>    3. Compute $\boldsymbol{\alpha}^{t+1}$ according to (17).<br>    4. Maintain active set $\mathcal{A}^t$ by (14).<br>    5. Maintain $\boldsymbol{w}_F(\boldsymbol{\alpha})$ according to (10).<br>**until** a non-drop step is performed. | **for** $i \in [n]$ **do**<br>    1. Find $f^*$ satisfying (18) for $i$-th sample.<br>    2. $\mathcal{A}_i^{s+1} = \mathcal{A}_i^s \cup \{f^*\}$.<br>    **for** $f \in \mathcal{A}_i$ **do**<br>        3.1 Update $\boldsymbol{\alpha}_f$ according to (19).<br>        3.2 Maintain $\boldsymbol{w}_F(\boldsymbol{\alpha})$ according to (10).<br>    **end for**<br>**end for** |

where $\mathcal{A}^t$ maintains an active set of bases of non-zero coefficients. Each iteration of AFW finds a direction $\boldsymbol{v}^+ := (\boldsymbol{v}_f^+)_{f \in \mathcal{F}}$ leading to the most descent amount according to the current gradient, subject to the simplex constraints:

$$\boldsymbol{v}_f^+ := \underset{\boldsymbol{v}_f \in \Delta^{|\mathcal{Y}_f|}}{argmin} \langle \nabla_{\boldsymbol{\alpha}_f} \mathcal{L}(\boldsymbol{\alpha}^t, \boldsymbol{\lambda}^t), \boldsymbol{v}_f \rangle = C(\boldsymbol{e}_{\bar{\boldsymbol{y}}_f} - \boldsymbol{e}_{\boldsymbol{y}_f^*}), \ \forall f \in \mathcal{F} \qquad (15)$$

where $\boldsymbol{y}_f^* := arg\max_{\boldsymbol{y}_f \in \mathcal{Y}_f \setminus \{\bar{\boldsymbol{y}}_f\}} \langle \nabla_{\boldsymbol{\alpha}_f} \mathcal{L}(\boldsymbol{\alpha}^t, \boldsymbol{\lambda}^t), \boldsymbol{e}_{\boldsymbol{y}_f} \rangle$ is the non-ground-truth labeling of factor $f$ of highest response. In addition, AFW finds the *away direction*

$$\boldsymbol{v}^- := \underset{\boldsymbol{v} \in \mathcal{A}^t}{argmax} \langle \nabla_{\boldsymbol{\alpha}} \mathcal{L}(\boldsymbol{\alpha}^t, \boldsymbol{\lambda}^t), \boldsymbol{v} \rangle, \qquad (16)$$

which corresponds to the basis that leads to the most descent amount when being removed. Then the update is determined by

$$\boldsymbol{\alpha}^{t+1} := \begin{cases} \boldsymbol{\alpha}^t + \gamma_F \boldsymbol{d}_F, & \langle \nabla_{\boldsymbol{\alpha}} \mathcal{L}, \boldsymbol{d}_F \rangle < \langle \nabla_{\boldsymbol{\alpha}} \mathcal{L}, \boldsymbol{d}_A \rangle \\ \boldsymbol{\alpha}^t + \gamma_A \boldsymbol{d}_A, & otherwise. \end{cases} \qquad (17)$$

where we choose between two descent directions $\boldsymbol{d}_F := \boldsymbol{v}^+ - \boldsymbol{\alpha}^t$ and $\boldsymbol{d}_A := \boldsymbol{\alpha}^t - \boldsymbol{v}^-$. The step size of each direction $\gamma_F := arg\min_{\gamma \in [0,1]} \mathcal{L}(\boldsymbol{\alpha}^t + \gamma \boldsymbol{d}_F)$ and $\gamma_A := arg\min_{\gamma \in [0, c_{\boldsymbol{v}^-}]} \mathcal{L}(\boldsymbol{\alpha}^t + \gamma \boldsymbol{d}_A)$ can be computed exactly due to the quadratic nature of (12). A step is called *drop step* if a step size $\gamma^* = c_{\boldsymbol{v}^-}$ is chosen, which leads to the removal of a basis $\boldsymbol{v}^-$ from the active set, and therefore the total number of drop steps can be bounded by half of the number of iterations $t$. Since a drop step could lead to insufficient descent, Algorithm 2 stops only if a *non-drop step* is performed. Note Algorithm 2 requires only a factorwise greedy search (15) instead of a structural maximization (2). In section 3.2 we show how the factorwise search can be implemented much more efficiently than structural ones. All the other steps (2-5) in Algorithm 2 can be computed in $O(|\mathcal{A}_f|nnz(\boldsymbol{\phi}_f))$, where $|\mathcal{A}_f|$ is the number of active states in factor $f$, which can be much smaller than $|\mathcal{Y}_f|$ when output domain is large.

In practice, a *Block-Coordinate Frank-Wolfe (BCFW)* method has much faster convergence than *Frank-Wolfe* method (Algorithm 2) [13, 9], but proving linear convergence for *BCFW* is also much more difficult [13], which prohibits its use in our analysis. In our implementation, however, we adopt the *BCFW* version since it turns out to be much more efficient. We include a detailed description on the BCFW version in Appendix-A (Algorithm 4).

**Large Number of Factors** Many structured prediction problems, such as alignment, segmentation, and multilabel prediction (Fig. 1, right), comprise binary variables and large number of factors with small domains, for which Algorithm 2 does not yield any computational advantage. For this type of problem, we minimize (12) via one pass of *Block-Greedy Coordinate Descent (BGCD)* (Algorithm 3) instead. Let $Q_{\max}$ be an upper bound on the eigenvalue of Hessian matrix of each block $\nabla^2_{\boldsymbol{\alpha}_f} \mathcal{L}(\boldsymbol{\alpha})$. For binary variables of pairwise factor, we have $Q_{\max} = 4(\max_{f \in F} \|\boldsymbol{\phi}_f\|^2 + 1)$. Each iteration of BGCD finds a factor that leads to the most progress

$$f^* := \underset{f \in \mathcal{F}(\boldsymbol{x}_i)}{argmin} \left( \min_{\boldsymbol{\alpha}_f + \boldsymbol{d} \in \Delta^{|\mathcal{Y}_f|}} \langle \nabla_{\boldsymbol{\alpha}_f} \mathcal{L}(\boldsymbol{\alpha}^t, \boldsymbol{\lambda}^t), \boldsymbol{d} \rangle + \frac{Q_{\max}}{2} \|\boldsymbol{d}\|^2 \right). \qquad (18)$$

for each instance $\boldsymbol{x}_i$, adds them into the set of active factors $\mathcal{A}_i$, and performs updates by solving block subproblems

$$\boldsymbol{d}_f^* = \underset{\boldsymbol{\alpha}_f + \boldsymbol{d} \in \Delta^{|\mathcal{Y}_f|}}{argmin} \langle \nabla_{\boldsymbol{\alpha}_f} \mathcal{L}(\boldsymbol{\alpha}^t, \boldsymbol{\lambda}^t), \boldsymbol{d} \rangle + \frac{Q_{\max}}{2} \|\boldsymbol{d}\|^2 \qquad (19)$$

for each factor $f \in \mathcal{A}_i$. Note $|\mathcal{A}_i|$ is bounded by the number of GDMM iterations and it converges to a constant much smaller than $|\mathcal{F}(\boldsymbol{x}_i)|$ in practice. We address in the next section how a joint FMO can be performed to compute (18) in time sublinear to $|\mathcal{F}(\boldsymbol{x}_i)|$ in the binary-variable case.

### 3.2  Greedy Search via Factorwise Maximization Oracle (FMO)

The main difference between the FMO and structural maximization oracle (2) is that the former involves only simple operations such as inner products or table look-ups for which one can easily come up with data structures or approximation schemes to lower the complexity. In this section, we present two approaches to realize sublinear-time FMOs for two types of factors widely used in practice. We will describe in terms of pairwise factors, but the approach can be naturally generalized to factors involving more variables.

**Indicator Factor**   Factors $\theta_f(\boldsymbol{x}_f, \boldsymbol{y}_f)$ of the form

$$\langle \boldsymbol{w}_F, \boldsymbol{\phi}_F(\boldsymbol{x}_f, \boldsymbol{y}_f)\rangle = v(\boldsymbol{x}_f, \boldsymbol{y}_f) \tag{20}$$

are widely used in practice. It subsumes the bigram factor $v(y_i, y_j)$ that is prevalent in sequence, grid, and network labeling problems, and also factors that map an input-output pair $(x, y)$ directly to a score $v(x, y)$. For this type of factor, one can maintain ordered multimaps for each factor template $F$, which support ordered visits of $\{v(\boldsymbol{x}, (y_i, y_j))\}_{(y_i, y_j) \in \mathcal{Y}_f}$, $\{v(\boldsymbol{x}, (y_i, y_j))\}_{y_j \in \mathcal{Y}_j}$ and $\{v(\boldsymbol{x}, (y_i, y_j))\}_{y_i \in \mathcal{Y}_i}$. Then to find $\boldsymbol{y}_f$ that maximizes (26), we compare the maximizers in 4 cases: (i) $(y_i, y_j) : m_{if}(y_i) = m_{jf}(y_j) = 0$, (ii) $(y_i, y_j) : m_{if}(y_i) = 0$, (iii) $(y_i, y_j) : m_{jf}(y_j) = 0$, (iv) $(y_i, y_j) : m_{jf}(y_j) \neq 0, m_{if}(y_i) \neq 0$. The maximization requires $O(|\mathcal{A}_i||\mathcal{A}_j|)$ in cases (ii)-(iv) and $O(\max(|\mathcal{A}_i||\mathcal{Y}_j|, |\mathcal{Y}_i||\mathcal{A}_j|))$ in case (i) (see details in Appendix C-1). However, in practice we observe an $O(1)$ cost for case (i) and the bottleneck is actually case (iv), which requires $O(|\mathcal{A}_i||\mathcal{A}_j|)$.

Note the ordered multimaps need maintenance whenever the vector $\boldsymbol{w}_F(\boldsymbol{\alpha})$ is changed. Fortunately, since the indicator factor has $v(\boldsymbol{y}_f, \boldsymbol{x}) = \sum_{f \in F, \boldsymbol{x}_f = \boldsymbol{x}} \alpha_f(\boldsymbol{y}_f)$, each update (25) leads to at most $|\mathcal{A}_f|$ changed elements, which gives a maintenance cost bounded by $O(|\mathcal{A}_f| \log(|\mathcal{Y}_F|))$. On the other hand, the space complexity is bounded by $O(|\mathcal{Y}_F||\mathcal{X}_F|)$ since the map is shared among factors.

**Binary-Variable Interaction Factor**   Many problems consider pairwise-interaction factor between binary variables, where the factor domain is small but the number of factors is large. For this type of problem, there is typically an rare outcome $y_f^A \in \mathcal{Y}_F$. We call factors exhibiting such outcome as *active factors* and the score of a labeling is determined by the score of the active factors (inactive factors give score 0). For example, in the problem of *multilabel prediction with pairwise interactions* (Fig. 1, right), an active unigram factor has outcome $y_j^A = 1$ and an active bigram factor has $\boldsymbol{y}_f^A = (1, 1)$, and each sample typically has only few outputs with value 1.

For this type of problem, we show that the gradient magnitude w.r.t. $\boldsymbol{\alpha}_f$ for a bigram factor $f$ can be determined by the gradient w.r.t. $\alpha_f(\boldsymbol{y}_f^A)$ when one of its incoming message $m_{jf}$ or $m_{if}$ is 0 (see details in Appendix C-2). Therefore, we can find the greedy factor (18) by maintaining an ordered multimap for the scores of outcome $\boldsymbol{y}_f^A$ in each factor $\{v(\boldsymbol{y}_f^A, \boldsymbol{x}_f)\}_{f \in F}$. The resulting complexity for finding a factor that maximizes (18) is then reduced from $O(|\mathcal{Y}_i||\mathcal{Y}_j|)$ to $O(|\mathcal{A}_i||\mathcal{A}_j|)$, where the latter is for comparison among factors that have both messages $m_{if}$ and $m_{jf}$ being non-zero.

**Inner-Product Factor**   We consider another widely-used type of factor of the form

$$\theta_f(\boldsymbol{x}_f, \boldsymbol{y}_f) = \langle \boldsymbol{w}_F, \boldsymbol{\phi}_F(\boldsymbol{x}_f, \boldsymbol{y}_f)\rangle = \langle \boldsymbol{w}_F(\boldsymbol{y}_f), \boldsymbol{\phi}_F(\boldsymbol{x}_f)\rangle$$

where all labels $\boldsymbol{y}_f \in \mathcal{Y}_f$ share the same feature mapping $\boldsymbol{\phi}_F(\boldsymbol{x}_f)$ but with different parameters $\boldsymbol{w}_F(\boldsymbol{y}_f)$. We propose a simple sampling approximation method with a performance guarantee for the convergence of GDMM. Note although one can apply similar sampling schemes to the structural maximization oracle (2), it is hard to guarantee the quality of approximation. The sampling method divides $\mathcal{Y}_f$ into $\nu$ mutually exclusive subsets $\mathcal{Y}_f = \bigcup_{k=1}^{\nu} \mathcal{Y}_f^{(k)}$, and realizes an approximate FMO by first sampling $k$ uniformly from $[\nu]$ and returning

$$\hat{\boldsymbol{y}}_f \in arg \max_{\boldsymbol{y}_f \in \mathcal{Y}_f^{(k)}} \langle \boldsymbol{w}_F(\boldsymbol{y}_f), \boldsymbol{\phi}_F(\boldsymbol{x}_f)\rangle. \tag{21}$$

Note there is at least $1/\nu$ probability that $\hat{\boldsymbol{y}}_f \in arg\max_{\boldsymbol{y}_f \in \mathcal{Y}_f} \langle \boldsymbol{w}_F(\boldsymbol{y}_f), \boldsymbol{\phi}_F(\boldsymbol{x}_f) \rangle$ since at least one partition $\mathcal{Y}_f^{(k)}$ contains a label of the highest score. In section 3.3, we show that this approximate FMO still ensures convergence with a rate scaled by $1/\nu$. In practice, since the set of active labels is not changing frequently during training, once an active label $\boldsymbol{y}_f$ is sampled, it will be kept in the active set $\mathcal{A}_f$ till the end of the algorithm and thus results in a convergence rate similar to that of an exact FMO. Note for problems of binary variables with large number of inner-product factors, the sampling technique applies similarly by simply partitioning factors as $\mathcal{F}_i = \bigcup_{k=1}^{\nu} \mathcal{F}_i^{(k)}$ and searching active factors only within one randomly chosen partition at a time.

### 3.3 Convergence Analysis

We show the iteration complexity of the GDMM algorithm with an $1/\nu$-approximated FMO given in section 3.2. The convergence guarantee for exact FMOs can be obtained by setting $\nu = 1$. The analysis leverages recent analysis on the global linear convergence of Frank-Wolfe variants [8] for function of the form (12) with a polyhedral domain, and also the analysis in [5] for Augmented Lagrangian based method. This type of greedy Augmented Lagrangian Method was also analyzed previously under different context [23, 24, 22].

Let $d(\boldsymbol{\lambda}) = \min_{\boldsymbol{\alpha}} \mathcal{L}(\boldsymbol{\alpha}, \boldsymbol{\lambda})$ be the dual objective of (12), and let

$$\Delta_d^t := d^* - d(\boldsymbol{\lambda}^t), \quad \Delta_p^t := \mathcal{L}(\boldsymbol{\alpha}^t, \boldsymbol{\lambda}^t) - d(\boldsymbol{\lambda}^t) \tag{22}$$

be the dual and primal suboptimality of problem (10) respectively. We have the following theorems.

**Theorem 1** (Convergence of GDMM with AFW). *The iterates $\{(\boldsymbol{\alpha}^t, \boldsymbol{\lambda}^t)\}_{t=1}^{\infty}$ produced by Algorithm 1 with step 1 performed by Algorithm 2 has*

$$E[\Delta_p^t + \Delta_d^t] \leq \epsilon \ for \ t \geq \omega \log(\frac{1}{\epsilon}) \tag{23}$$

*for any $0 < \eta \leq \frac{\rho}{4 + 16(1+\nu)mQ/\mu_{\mathcal{M}}}$ with $\omega = \max\left\{2(1 + 4\frac{mQ(1+\nu)}{\mu_{\mathcal{M}}}), \frac{\tau}{\eta}\right\}$, where $\mu_{\mathcal{M}}$ is the generalized geometric strong convexity constant of (12), $Q$ is the Lipschitz-continuous constant for the gradient of objective (12), and $\tau > 0$ is a constant depending on optimal solution set.*

**Theorem 2** (Convergence of GDMM with BGCD). *The iterates $\{(\boldsymbol{\alpha}^t, \boldsymbol{\lambda}^t)\}_{t=1}^{\infty}$ produced by Algorithm 1 with step 1 performed by Algorithm 3 has*

$$E[\Delta_p^t + \Delta_d^t] \leq \epsilon \ for \ t \geq \omega_1 \log(\frac{1}{\epsilon}) \tag{24}$$

*for any $0 < \eta \leq \frac{\rho}{4(1+Q_{\max}\nu/\mu_1)}$ with $\omega_1 = \max\left\{2(1 + \frac{Q_{\max}\nu}{\mu_1}), \frac{\tau}{\eta}\right\}$, where $\mu_1$ is the generalized strong convexity constant of objective (12) and $Q_{\max} = \max_{f \in \mathcal{F}} Q_f$ is the factorwise Lipschitz-continuous constant on the gradient.*

## 4 Experiments

In this section, we compare with existing approaches on Sequence Labeling and Multi-label prediction with pairwise interaction. The algorithms in comparison are: (i) *BCFW*: a Block-Coordinate Frank-Wolfe method based on structural oracle [9], which outperforms other competitors such as Cutting-Plane, FW, and online-EG methods in [9]. (ii) *SSG*: an implementation of the Stochastic Subgradient method [16]. (iii) *Soft-BCFW*: Algorithm proposed in ([12]), which avoids structural oracle by minimizing an approximate objective, where a parameter $\rho$ controls the precision of the approximation. We tuned the parameter and chose two of the best on the figure. For BCFW and SSG, we adapted the MATLAB implementation provided by authors of [9] into C++, which is an order of magnitude faster. All other implementations are also in C++. The results are compared in terms of primal objective (achieved by $\boldsymbol{w}$) and test accuracy.

Our experiments are conducted on 4 public datasets: *POS*, *ChineseOCR*, *RCV1-regions*, and *EUR-Lex* (directory codes). For sequence labeling we experiment on *POS* and *ChineseOCR*. The *POS* dataset is a subset of Penn treebank[2] that contains 3,808 sentences, 196,223 words, and 45 POS labels. The HIT-MW[3] *ChineseOCR* dataset is a hand-written Chinese character dataset from [17]. The

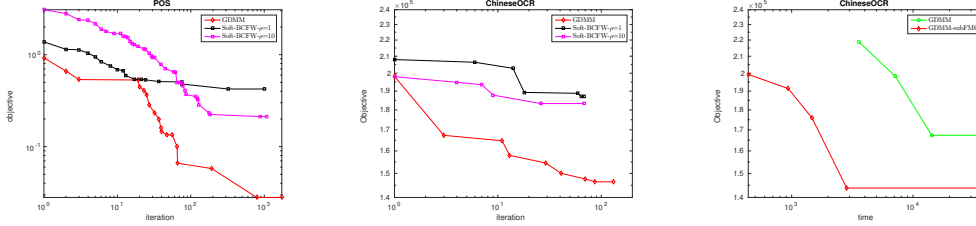

Figure 2: (left) Compare two FMO-based algorithms (GDMM, Soft-BCFW) in number of iterations. (right) Improvement in training time given by sublinear-time FMO.

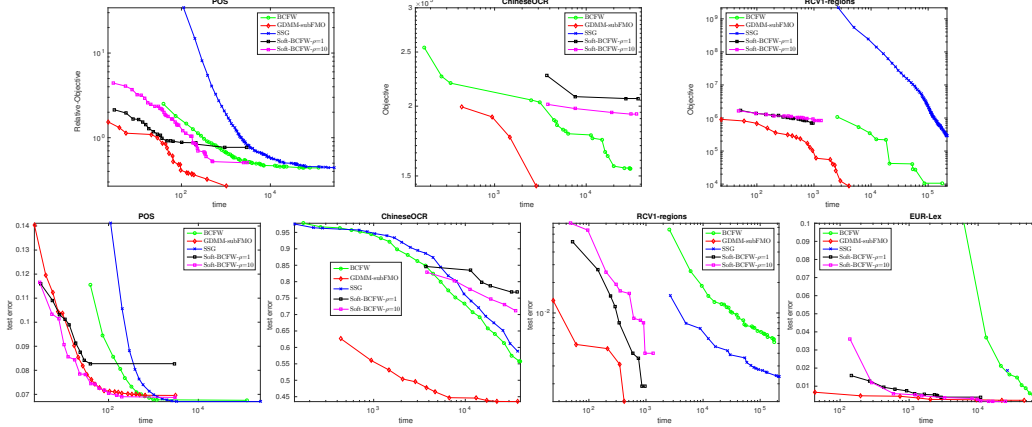

Figure 3: Primal Objective v.s. Time and Test error v.s. Time plots. Note that figures of objective have showed that SSG converges to a objective value much higher than all other methods, this is also observed in [9]. Note the training objective for the EUR-Lex data set is too expensive to compute and we are unable to plot the figure.

dataset has 12,064 hand-written sentences, and a total of 174,074 characters. The vocabulary (label) size is 3,039. For the Correlated Multilabel Prediction problems, we experiment on two benchmark datasets *RCV1-regions*[4] and *EUR-Lex* (directory codes)[5]. The *RCV1-regions* dataset has 228 labels, 23,149 training instances and 47,236 features. Note that a smaller version of *RCV1* with only 30 labels and 6000 instances is used in [11, 12]. *EUR-Lex* (directory codes) has 410 directory codes as labels with a sample size of 19,348. We first compare GDMM (without subFMO) with Soft-BCFW in Figure 2. Due to the approximation (controlled by $\rho$), Soft-BCFW can converge to a suboptimal primal objective value. While the gap decreases as $\rho$ increases, its convergence becomes also slower. GDMM, on the other hand, enjoys a faster convergence. The sublinear-time implementation of FMO also reduces the training time by an order of magnitude on the ChineseOCR data set, as showed in Figure 2 (right). More general experiments are showed in Figure 3. When the size of output domain is small (POS dataset), GDMM-subFMO is competitive to other solvers. As the size of output domain grows (ChineseOCR, RCV1, EUR-Lex), the complexity of structural maximization oracle grows linearly or even quadratically, while the complexity of GDMM-subFMO only grows sublinearly in the experiments. Therefore, GDMM-subFMO achieves orders-of-magnitude speedup over other methods. In particular, when running on ChineseOCR and EUR-Lex, each iteration of SSG, GDMM, BCFW and Soft-BCFW take over $10^3$ seconds, while it only takes a few seconds in GDMM-subFMO.

**Acknowledgements.** We acknowledge the support of ARO via W911NF-12-1-0390, NSF via grants CCF-1320746, CCF-1117055, IIS-1149803, IIS-1546452, IIS-1320894, IIS-1447574, IIS-1546459, CCF-1564000, DMS-1264033, and NIH via R01 GM117594-01 as part of the Joint DMS/NIGMS Initiative to Support Research at the Interface of the Biological and Mathematical Sciences.

## Footnotes

[1]$\boldsymbol{\alpha}_j$ is also dual variables for responses on unigram factors. We define $\mathcal{U} := \mathcal{V}$ and $\boldsymbol{\alpha}_f := \boldsymbol{\alpha}_j, \forall f \in \mathcal{U}$.

[2] https://catalog.ldc.upenn.edu/LDC99T42

[3] https://sites.google.com/site/hitmwdb/

[4] www.csie.ntu.edu.tw/~cjlin/libsvmtools/datasets/multilabel.html

[5] mulan.sourceforge.net/datasets-mlc.html

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
