[Supplementary Material · DDL-Supplementary.pdf]

# 5 Appendix A: Block-Coordinate Frank-Wolfe (BCFW)— Practically Faster-Convergent Variant

---

**Algorithm 4** Block-Coordinate Frank Wolfe (improving upon Algorithm 2)

---
  **for** $s = 1$ to $|\mathcal{F}|$ **do**
    1. Draw $f \in \mathcal{F}$ uniformly at random.
    2. Find a greedy direction $\boldsymbol{v}_f^+$ satisfying (15).
    3. $\mathcal{A}_f^{s+1} = \mathcal{A}_f^s \cup \{\boldsymbol{v}_f^+\}$.
    4. Solve (25) with active set $\mathcal{A}_f^{s+1}$.
    5. Maintain $\boldsymbol{w}_F(\boldsymbol{\alpha})$.
  **end for**

---

The *Block-Coordinate Frank-Wolfe (BCFW)* (Algorithm 4) differs from *Frank-Wolfe* (Algorithm 2) in that it updates dual variables $\boldsymbol{\alpha}_f$ of each factor sequentially, and the bases $\boldsymbol{v}_f$ and active sets $\mathcal{A}_f$ are maintained for each factor $f$ separately. For each iteration of BCFW, we find a greedy direction $\boldsymbol{v}_f^+$ in the same way (15) as AFW, but for one factor at a time. Then we add $\boldsymbol{v}_f^+$ to an active set $\mathcal{A}_f$ maintained for each factor. Since in BCFW we update one factor at a time, we can minimize the following *block active-set subproblem*

$$\boldsymbol{d}_{\mathcal{A}_f}^* = \underset{\boldsymbol{\alpha}_f + \boldsymbol{d}_{A_f} \in \Delta^{|\mathcal{Y}_f|}}{argmin} \quad \langle \nabla_{\boldsymbol{\alpha}_f} \mathcal{L}(\boldsymbol{\alpha}, \boldsymbol{\lambda}^t), \boldsymbol{d}_{A_f} \rangle + \frac{Q_f}{2} \|\boldsymbol{d}_{A_f}\|^2 \tag{25}$$

where $Q_f$ is an upper bound on the Hessian of variables in the active set (discussed in section 5.1). The active-set subproblem (25) can be solved via a simplex projection in time $O(|\mathcal{A}_f|)$ [2]. Furthermore, by maintaining $\boldsymbol{w}_F(\boldsymbol{\alpha})$ after solving each sub-problem (25), we can compute the gradient

$$\nabla_{\alpha_f(\boldsymbol{y}_f)} \mathcal{L} = \langle \boldsymbol{w}_F, \boldsymbol{\phi}_f(\boldsymbol{x}_f, \boldsymbol{y}_f) \rangle - \delta_f(\boldsymbol{y}_f, \bar{\boldsymbol{y}}_f) + \rho_f \sum_{j \in \mathcal{N}(f), \, y_j = [\boldsymbol{y}_f]_j} m_{jf}(y_j) \tag{26}$$

for $\boldsymbol{y}_f \in \mathcal{A}_f$ in time $O(|\mathcal{A}_f| nnz(\boldsymbol{\phi}_f))$, where $\rho_f = -\rho$, $\delta_f = \delta_j$ for $f \in \mathcal{U}$ and $\rho_f = \rho$, $\delta_f = 0$ for $f \notin \mathcal{U}$. Note the size of active set $|\mathcal{A}_f|$ is bounded by the number of GDMM iterations, and in practice $|\mathcal{A}_f|$ converges to a constant much smaller than $|\mathcal{Y}_f|$ for problems of large output domains. Therefore, the bottleneck of the BCFW algorithm lies in the step (15), which as we show in Section 3.2, can be computed in time sublinear to $|\mathcal{Y}_f|$ via an efficient FMO.

## 5.1 Constant $Q_f$ in Problem (25)

The constant $Q_f$ is an upper bound on the maximum eigenvalue of the Hessian submatrix for variables in the active set $\mathcal{A}_f$, that is, $\|[\Phi_f \Phi_f^T]_{\mathcal{A}_f}\| + \rho \sum_{j \in \mathcal{N} f} \|[M_{jf}^T M_{jf}]_{\mathcal{A}_f}\|$, where the notation $[.]_{\mathcal{A}}$ denotes the sub-matrix formed by row and column indexes in $\mathcal{A}$ and $\|.\|$ is the spectral norm of a matrix.

For many types of factors used in practice, $Q_f = O(|\mathcal{A}_f|)$ and is easy to compute in the beginning.

In particular, for unigram factor, we have $M_{jf} = I$ and thus $\|[M_{jf}^T M_{jf}]_{\mathcal{A}}\| = 1$, and for higher-order factor we have $\|[M_{jf}^T M_{jf}]_{\mathcal{A}}\| = |\mathcal{A}|$.

As for the term $\|[\Phi_f \Phi_f^T]_{\mathcal{A}}\|$. In most of applications, $\Phi_f$ is a $|\mathcal{Y}_f| \times (|\mathcal{Y}_f|d)$ block-diagonal matrix that duplicates $1 \times d$ feature vector $\boldsymbol{\phi}_f(\boldsymbol{x}_f)^T$ for $|\mathcal{Y}_f|$ times, for which we have $\|[\Phi_f \Phi_f^T]_{\mathcal{A}}\| = \|\boldsymbol{\phi}_f(\boldsymbol{x}_f)\|^2$. Note in this case, the quadratic upper bound in (25) is tight for unigram factors.

# 6 Appendix B: Convergence of GDMM

## 6.1 Proof of Theorem (1)

Recall that the Augmented Lagrangian $\mathcal{L}(\boldsymbol{\alpha}, \boldsymbol{\lambda})$ is of the form

$$\mathcal{L}(\boldsymbol{\alpha}, \boldsymbol{\lambda}) := G(\boldsymbol{\alpha}) + \langle \boldsymbol{\lambda}, M\boldsymbol{\alpha} \rangle + \frac{\rho}{2}\|M\boldsymbol{\alpha}\|^2.$$

where $M$ is "the number of consistency constraints" by "the number of variables" matrix and $M\boldsymbol{\alpha} = \mathbf{0}$ encodes all constraints of the form

$$M_{jf}\boldsymbol{\alpha}_f - \boldsymbol{\alpha}_j = [\ M_{jf} \quad -I_j\ ] \begin{bmatrix} \boldsymbol{\alpha}_f \\ \boldsymbol{\alpha}_j \end{bmatrix} = \mathbf{0}.$$

The function

$$G(\boldsymbol{\alpha}) = \frac{1}{2} \sum_{F \in \mathcal{F}} \|\boldsymbol{w}_F(\boldsymbol{\alpha}_F)\|^2 - \sum_{j \in \mathcal{U}} \boldsymbol{\delta}_j^T \boldsymbol{\alpha}_j$$

can be written in a compact form as

$$\begin{aligned} G(\boldsymbol{\alpha}) &= \frac{1}{2}\|\boldsymbol{w}(\boldsymbol{\alpha})\|^2 + \boldsymbol{\delta}^T\boldsymbol{\alpha} \\ &= \frac{1}{2}\|\Phi^T\boldsymbol{\alpha}\|^2 + \boldsymbol{\delta}^T\boldsymbol{\alpha} \end{aligned} \tag{27}$$

where $\Phi$ is the "number of variables (in $\boldsymbol{\alpha}$)" by "number of parameters (in $\boldsymbol{w}$)" design matrix.

Now let $\boldsymbol{\alpha}$ be the "primal variables" and denote

$$\boldsymbol{\alpha}(\boldsymbol{\lambda}) := \{\boldsymbol{\alpha}|\boldsymbol{\alpha} = arg\min_{\boldsymbol{\alpha}}\ \mathcal{L}(\boldsymbol{\alpha}, \boldsymbol{\lambda})\} \tag{28}$$

with

$$\bar{\boldsymbol{\alpha}}^t := \underset{\bar{\boldsymbol{\alpha}} \in \boldsymbol{\alpha}(\boldsymbol{\lambda}^t)}{argmin}\|\bar{\boldsymbol{\alpha}} - \boldsymbol{\alpha}^t\|,$$

and let $\mathcal{M} = \{\boldsymbol{\alpha} \mid \boldsymbol{\alpha}_f \in \Delta^{|\mathcal{Y}_f|}, \forall f \in \mathcal{F}\}$. The dual objective of the augmented problem is

$$d(\boldsymbol{\lambda}) = \min_{\boldsymbol{\alpha} \in \mathcal{M}} \mathcal{L}(\boldsymbol{\alpha}, \boldsymbol{\lambda})$$

and

$$d^* = \max_{\boldsymbol{\lambda}} d(\boldsymbol{\lambda})$$

is the optimal dual objective value.

Then we measure the sub-optimality of iterates $\{(\boldsymbol{\alpha}^t, \boldsymbol{\lambda}^t)\}_{t=1}^T$ given by GDMM in terms of dual function difference

$$\Delta_d^t = d^* - d(\boldsymbol{\lambda}^t)$$

and the primal function difference for a given dual iterate $\boldsymbol{\lambda}^t$:

$$\Delta_p^t = \mathcal{L}(\boldsymbol{\alpha}^{t+1}, \boldsymbol{\lambda}^t) - d(\boldsymbol{\lambda}^t)$$

yielded by $\boldsymbol{\alpha}^{t+1}$ obtained from one pass of FC-BCFW algorithm on $\boldsymbol{\alpha}$. Then we have following lemma.

**Lemma 1** (Dual Progress). *Each iteration of GDMM (Algorithm 1) has*

$$\Delta_d^t - \Delta_d^{t-1} \leq -\eta(M\boldsymbol{\alpha}^t)^T(M\bar{\boldsymbol{\alpha}}^t). \tag{29}$$

*Proof.*

$$\begin{aligned} \Delta_d^t - \Delta_d^{t-1} &= d^* - d(\boldsymbol{\lambda}^t) - d^* - d(\boldsymbol{\lambda}^{t-1}) \\ &= \mathcal{L}(\bar{\boldsymbol{\alpha}}^{t-1}, \boldsymbol{\lambda}^{t-1}) - \mathcal{L}(\bar{\boldsymbol{\alpha}}^t, \boldsymbol{\lambda}^t) \\ &\leq \mathcal{L}(\bar{\boldsymbol{\alpha}}^t, \boldsymbol{\lambda}^{t-1}) - \mathcal{L}(\bar{\boldsymbol{\alpha}}^t, \boldsymbol{\lambda}^t) \\ &= \langle \boldsymbol{\lambda}^{t-1} - \boldsymbol{\lambda}^t, M\bar{\boldsymbol{\alpha}}^t \rangle \\ &= -\eta \langle M\boldsymbol{\alpha}^t, M\bar{\boldsymbol{\alpha}}^t \rangle \end{aligned}$$

where the first inequality follows the optimality of $\bar{\boldsymbol{\alpha}}^{t-1}$ for the function $\mathcal{L}(\boldsymbol{\alpha}, \boldsymbol{\lambda}^{t-1})$ defined by $\boldsymbol{\lambda}^{t-1}$, and the last equality follows the dual update in GDMM (13). $\qquad\square$

On the other hand, the following lemma gives an expression on the primal progress that is independent of the algorithm used for minimizing Augmented Lagrangian

**Lemma 2** (Primal Progress). *Each iteration of GDMM (Algorithm 1) has*

$$\Delta_p^t - \Delta_p^{t-1} \leq \mathcal{L}(\boldsymbol{\alpha}^{t+1}, \boldsymbol{\lambda}^t) - \mathcal{L}(\boldsymbol{\alpha}^t, \boldsymbol{\lambda}^t)$$
$$+ \eta \|M\boldsymbol{\alpha}^t\|^2 - \eta \langle M\boldsymbol{\alpha}^t, M\bar{\boldsymbol{\alpha}}^t \rangle$$

*Proof.*

$$\Delta_p^t - \Delta_p^{t-1}$$
$$= \mathcal{L}(\boldsymbol{\alpha}^{t+1}, \boldsymbol{\lambda}^t) - \mathcal{L}(\boldsymbol{\alpha}^t, \boldsymbol{\lambda}^{t-1}) - (d(\boldsymbol{\lambda}^t) - d(\boldsymbol{\lambda}^{t-1}))$$
$$\leq \mathcal{L}(\boldsymbol{\alpha}^{t+1}, \boldsymbol{\lambda}^t) - \mathcal{L}(\boldsymbol{\alpha}^t, \boldsymbol{\lambda}^t) + \mathcal{L}(\boldsymbol{\alpha}^t, \boldsymbol{\lambda}^t) - \mathcal{L}(\boldsymbol{\alpha}^t, \boldsymbol{\lambda}^{t-1}) + (d(\boldsymbol{\lambda}^{t-1}) - d(\boldsymbol{\lambda}^t))$$
$$\leq \mathcal{L}(\boldsymbol{\alpha}^{t+1}, \boldsymbol{\lambda}^t) - \mathcal{L}(\boldsymbol{\alpha}^t, \boldsymbol{\lambda}^t) + \eta \|M\boldsymbol{\alpha}^t\|^2 - \eta \langle M\boldsymbol{\alpha}^t, M\bar{\boldsymbol{\alpha}}^t \rangle$$

where the last inequality uses Lemma 1 on $d(\boldsymbol{\lambda}^{t-1}) - d(\boldsymbol{\lambda}^t) = \Delta_d^t - \Delta_d^{t-1}$. $\square$

By combining results of Lemma 1 and 2, we can obtain a joint progress of the form

$$\Delta_d^t - \Delta_d^{t-1} + \Delta_p^t - \Delta_p^{t-1}$$
$$\leq \mathcal{L}(\boldsymbol{\alpha}^{t+1}, \boldsymbol{\lambda}^t) - \mathcal{L}(\boldsymbol{\alpha}^t, \boldsymbol{\lambda}^t) + \eta \|M\boldsymbol{\alpha}^t - M\bar{\boldsymbol{\alpha}}^t\|^2 - \eta \|M\bar{\boldsymbol{\alpha}}^t\|^2 \tag{30}$$

Note the only positive term in (30) is the second one. To guarantee the descent of joint progress, we upper bound the three terms in (30) with the following lemmas.

**Lemma 3.**

$$\|M\boldsymbol{\alpha}^t - M\bar{\boldsymbol{\alpha}}^t\|^2 \leq \frac{2}{\rho}(\mathcal{L}(\boldsymbol{\alpha}^t, \boldsymbol{\lambda}^t) - \mathcal{L}(\bar{\boldsymbol{\alpha}}^t, \boldsymbol{\lambda}^t)) \tag{31}$$

*Proof.* Let

$$\tilde{\mathcal{L}}(\boldsymbol{\alpha}, \boldsymbol{\lambda}) = h(\boldsymbol{\alpha}) + \frac{\rho}{2}\|M\boldsymbol{\alpha}\|^2,$$

where

$$h(\boldsymbol{\alpha}) = G(\boldsymbol{\alpha}) + \langle \boldsymbol{\lambda}, M\boldsymbol{\alpha} \rangle + \boldsymbol{I}_{\boldsymbol{\alpha} \in \mathcal{M}}.$$

, $\boldsymbol{I}_{\boldsymbol{\alpha} \in \mathcal{M}} = 0$ if $\boldsymbol{\alpha} \in \mathcal{M}$ and $\boldsymbol{I}_{\boldsymbol{\alpha} \in \mathcal{M}} = \infty$ otherwise. Note we have $\tilde{\mathcal{L}}(\bar{\boldsymbol{\alpha}}^t, \boldsymbol{\lambda}^t) = \mathcal{L}(\bar{\boldsymbol{\alpha}}^t, \boldsymbol{\lambda}^t)$ and $\tilde{\mathcal{L}}(\boldsymbol{\alpha}^t, \boldsymbol{\lambda}^t) = \mathcal{L}(\boldsymbol{\alpha}^t, \boldsymbol{\lambda}^t)$ due to feasible iterates. Due to the optimality of $\bar{\boldsymbol{\alpha}}^t$, we have

$$\mathbf{0} = \boldsymbol{\sigma} + M^T M\bar{\boldsymbol{\alpha}}^t \in \partial_{\boldsymbol{\alpha}} \tilde{\mathcal{L}}(\bar{\boldsymbol{\alpha}}^t, \boldsymbol{\lambda})$$

for some $\boldsymbol{\sigma} \in \partial h(\bar{\boldsymbol{\alpha}}^t)$. And by the convexity of $h(\cdot)$ and the strong convexity of $\frac{\rho}{2}\|\cdot\|^2$, we have

$$h(\boldsymbol{\alpha}^t) - h(\bar{\boldsymbol{\alpha}}^t) \geq \langle \boldsymbol{\sigma}, \boldsymbol{\alpha}^t - \bar{\boldsymbol{\alpha}}^t \rangle$$

and

$$\frac{\rho}{2}\|M\boldsymbol{\alpha}^t\|^2 - \frac{\rho}{2}\|M\bar{\boldsymbol{\alpha}}^t\|^2 \geq \langle M^T M\bar{\boldsymbol{\alpha}}^t, \boldsymbol{\alpha}^t - \bar{\boldsymbol{\alpha}}^t \rangle + \frac{\rho}{2}\|M\boldsymbol{\alpha}^t - M\bar{\boldsymbol{\alpha}}^t\|^2$$

Then the above two together imply

$$\mathcal{L}(\boldsymbol{\alpha}^t, \boldsymbol{\lambda}^t) - \mathcal{L}(\bar{\boldsymbol{\alpha}}^t, \boldsymbol{\lambda}^t) \geq \frac{\rho}{2}\|M(\boldsymbol{\alpha}^t) - M(\bar{\boldsymbol{\alpha}}^t)\|^2$$

which leads to our conclusion. $\square$

**Lemma 4** (*Hong and Luo 2012*). *There is a constant $\tau > 0$ such that*

$$\Delta_d(\boldsymbol{\lambda}) \leq \tau \|M\bar{\boldsymbol{\alpha}}(\boldsymbol{\lambda})\|^2. \tag{32}$$

*for any $\boldsymbol{\lambda}$ and any minimizer $\bar{\boldsymbol{\alpha}}(\boldsymbol{\lambda})$ satisfying* (28).

*Proof.* This is a lemma adapted from [5]. Since our objective (12) satisfies the *assumptions A(a)—A(e)* and *A(g)* in [5]. Then Lemma 3.1 of [5] guarantees that, as long as $\|\nabla d(\boldsymbol{\lambda})\|$ is bounded, there is a constant $\tau > 0$ s.t.

$$\Delta_d(\boldsymbol{\lambda}) \leq \tau \|\nabla d(\boldsymbol{\lambda})\|^2 = \|M\bar{\boldsymbol{\alpha}}(\boldsymbol{\lambda})\|^2$$

for all $\boldsymbol{\lambda}$. Note our problem satisfies the condition of bounded gradient magnitude since

$$\|\nabla d(\boldsymbol{\lambda})\| = \|M\bar{\boldsymbol{\alpha}}(\boldsymbol{\lambda})\| \leq \|M\bar{\boldsymbol{\alpha}}(\boldsymbol{\lambda})\|_1 \leq \|M\|_1 \|\bar{\boldsymbol{\alpha}}(\boldsymbol{\lambda})\|_1 \leq (\max_f |\mathcal{Y}_f|)|\mathcal{F}|$$

where the last inequality is because $\bar{\boldsymbol{\alpha}}(\boldsymbol{\lambda})$ lies in a simplex domain. $\square$

The remaining thing is to show that one pass of AFW (Algorithm 2) or BGCD (Algorithm 3) suffices to give a descent amount $\mathcal{L}(\boldsymbol{\alpha}^{t+1}, \boldsymbol{\lambda}^t) - \mathcal{L}(\boldsymbol{\alpha}^t, \boldsymbol{\lambda}^t)$ lower bounded by some constant multiple of the primal sub-optimality $\mathcal{L}(\boldsymbol{\alpha}^t, \boldsymbol{\lambda}^t) - \mathcal{L}(\bar{\boldsymbol{\alpha}}^t, \boldsymbol{\lambda}^t)$. If it is true, then by selecting a small enough GDMM step size $\eta$, the RHS of (30) would be negative. For AFW (Algorithm 2), this can be achieved by leveraging recent results from [8], which shows linear convergence of AFW, even for non-strongly convex function of the form (34). We thus have the following lemma.

**Lemma 5.** *The descent amount of Augmented Lagrangian function produced by one pass of AFW (Algorithm 2) (and FMO parameter $\nu$) has*

$$E[\mathcal{L}(\boldsymbol{\alpha}^{t+1}, \boldsymbol{\lambda}^t)] - \mathcal{L}(\boldsymbol{\alpha}^t, \boldsymbol{\lambda}^t) \leq -\frac{\mu_{\mathcal{M}}}{4(1+\nu)mQ}(\mathcal{L}(\boldsymbol{\alpha}^t, \boldsymbol{\lambda}^t) - \mathcal{L}(\bar{\boldsymbol{\alpha}}^t, \boldsymbol{\lambda}^t)) \tag{33}$$

*where $\mu_{\mathcal{M}}$ is the generalized geometric strong convexity constant for function $\mathcal{L}(\boldsymbol{\alpha})$ in domain $\mathcal{M}$, $Q$ is the Lipschitz-continuous constant of $\nabla_{\boldsymbol{\alpha}}\mathcal{L}(\boldsymbol{\alpha})$ and $m = |\mathcal{F}|$.*

*Proof.* Note the Augmented Lagrangian is of the form

$$H(\boldsymbol{\alpha}) := \mathcal{L}(\boldsymbol{\alpha}, \boldsymbol{\lambda}^t) = g(B\boldsymbol{\alpha}) + \langle \boldsymbol{b}, \boldsymbol{\alpha} \rangle \tag{34}$$

where

$$B := \begin{bmatrix} \Phi^T \\ M \end{bmatrix}, \quad \boldsymbol{b} := \boldsymbol{\delta} + M^T \boldsymbol{\lambda}^t$$

and function $g\left(\begin{bmatrix} \boldsymbol{w} \\ \boldsymbol{v} \end{bmatrix}\right) = \frac{1}{2}\|\boldsymbol{w}\|^2 + \frac{\rho}{2}\|\boldsymbol{v}\|^2 + const.$ is strongly convex with parameter $\bar{\rho} = \min(1, \rho)$. Without loss of generality, assume $\rho \leq 1$ and thus $\bar{\rho} = \rho$. Since we are minimizing the function subject to a convex, polyhedral domain $\mathcal{M}$, by Theorem 10 of [8], we have the *generalized geometrical strong convexity* constant $\mu_{\mathcal{M}}$ of the form

$$\mu_{\mathcal{M}} := \mu(PWidth(\mathcal{M}))^2 \tag{35}$$

where $PWidth(\mathcal{M}) > 0$ is the pyramidal width of the simplex domain $\mathcal{M}$ and $\mu$ is the *generalized strong convexity* constant of function (34) (defined by Lemma 9 of [8]). By definition of the geometric strong convexity constant, we have

$$H(\boldsymbol{\alpha}^t) - H^* \leq \frac{g_t^2}{2\mu_{\mathcal{M}}} \tag{36}$$

from (23) in [8], where $g_t := \langle -\nabla H(\boldsymbol{\alpha}^t), \boldsymbol{v}^F - \boldsymbol{v}^A \rangle$, and $\boldsymbol{v}^F$ is the Frank-Wolfe direction

$$\boldsymbol{v}^F := arg \min_{\boldsymbol{v} \in \mathcal{M}} \langle \nabla H(\boldsymbol{\alpha}^t), \boldsymbol{v} \rangle,$$

$\boldsymbol{v}^A$ is the away direction

$$\boldsymbol{v}^A := arg \max_{\boldsymbol{v} \in \mathcal{A}^t} \langle \nabla H(\boldsymbol{\alpha}^t), \boldsymbol{v} \rangle$$

Then let $m = |\mathcal{F}|$ be the number of factors. The FMO returns $\boldsymbol{v}_f^+ = \boldsymbol{v}_f^F$ with probability at least $\frac{1}{\nu}$, and suppose we set $\boldsymbol{v}_f^+$ to $\boldsymbol{\alpha}_f^t$ whenever $\langle \nabla_{\boldsymbol{\alpha}_f} H, \boldsymbol{v}_f^+ - \boldsymbol{\alpha}_f^t \rangle \not\leq 0$. We have $\langle \nabla H, \boldsymbol{d}_F \rangle \leq \frac{1}{\nu} \langle \nabla H, \boldsymbol{v}^F - \boldsymbol{\alpha}^t \rangle$ and thus

$$(1 + \frac{1}{\nu}) \langle \nabla H, \boldsymbol{d}^t \rangle \leq \frac{1}{\nu} \langle \nabla H, \boldsymbol{v}^F - \boldsymbol{\alpha}^t \rangle + \frac{1}{\nu} \langle \nabla H, \boldsymbol{\alpha}^t - \boldsymbol{v}^A \rangle$$

and $\langle \nabla H, \boldsymbol{d}^t \rangle \leq -\frac{1}{1+\nu} g_t$. Therefore, for any $\forall \gamma \in [0,1]$,

$$E[H(\boldsymbol{\alpha}^{t+1})] - H(\boldsymbol{\alpha}^t) \leq -\gamma \frac{g_t}{1+\nu} + \frac{Q}{2}\|\gamma(\boldsymbol{\alpha}^{t+1} - \boldsymbol{\alpha}^t)\|^2 \leq -\gamma \frac{g_t}{1+\nu} + \frac{2mQ}{2}\gamma^2 \qquad (37)$$

where $Q$ is an upper bound on the spectral norm of Hessian $\|\nabla^2 H(\boldsymbol{\alpha})\|$ and $2m$ is the square of the radius of domain $\mathcal{M}$. Now we need to consider two cases. When the greedy direction $\boldsymbol{d}_F$ in (17) is chosen, we have $\gamma^* = \min(\frac{g_s}{mQ_{\max}}, 1)$, which gives us

$$E[H(\boldsymbol{\alpha}^{t+1})] - H(\boldsymbol{\alpha}^t) \leq -\frac{g_t^2}{4(1+\nu)mQ}. \qquad (38)$$

While in case $\boldsymbol{d}_A$ in (17) is chosen, we have $\gamma^* = \min(\frac{g_s}{mQ_{\max}}, c_{\boldsymbol{v}^-})$. When $\gamma^* = c_{\boldsymbol{v}^-}$, a basis $\boldsymbol{v}^-$ is removed from the active set and this is called a *drop step* [8] and it is hard to show sufficient descent in this case. Nevertheless, we can ignore those drop steps since the number of them is at most half of the iterates. For a non-drop step $t$, with the error bound (36), we have

$$E[H(\boldsymbol{\alpha}^{t+1})] - H(\boldsymbol{\alpha}^t) \leq -\frac{\mu_{\mathcal{M}}(H(\boldsymbol{\alpha}^t) - H^*)}{4(1+\nu)mQ}. \qquad (39)$$

$\square$

The above Lemma shows a significant progress made by the AFW algorithm. In the following, we provide a similar Lemma for minimizing AL subproblem with the Block-Greedy Coordinate Descent (BGCD) (Algorithm 3). Note that for problem of the form (34), the optimal solution is profiled by a polyhedral set $\mathcal{S} := \{\boldsymbol{\alpha} \mid B\boldsymbol{\alpha} = \boldsymbol{t}^*, \boldsymbol{b}^T\boldsymbol{\alpha} = s^*, \boldsymbol{\alpha} \in \mathcal{M}\}$. Therefore, let $\bar{\boldsymbol{\alpha}} := \Pi_{\mathcal{S}}(\boldsymbol{\alpha})$. We can bound the distance of any feasible point $\boldsymbol{\alpha} \in \mathcal{M}$ to its projection $\Pi_{\mathcal{S}}(\boldsymbol{\alpha})$ on $\mathcal{S}$ using the Hoffman's inequality [4]

$$\|\bar{\boldsymbol{\alpha}} - \boldsymbol{\alpha}\|_{2,1}^2 = \sum_{i=1}^n (\sum_{f \in \mathcal{F}_i} \|\bar{\boldsymbol{\alpha}}_f - \boldsymbol{\alpha}_f\|_2)^2 \leq \theta_1 \left( \|B\boldsymbol{\alpha} - \boldsymbol{t}^*\|^2 + \|\boldsymbol{b}^T\boldsymbol{\alpha} - s^*\|^2 \right) \qquad (40)$$

where $\theta_1$ is a constant depending on the set $\mathcal{S}$. Then we can establish the following Lemma using the error bound (40).

**Lemma 6.** *The descent amount of Augmented Lagrangian function given by one pass of BGCD (Algorithm 3) with FMO multiplicative-approximation parameter $\nu$ has*

$$E[\mathcal{L}(\boldsymbol{\alpha}^{t+1}, \boldsymbol{\lambda}^t)] - \mathcal{L}(\boldsymbol{\alpha}^t, \boldsymbol{\lambda}^t) \leq \frac{-1}{1 + \nu Q_{\max}/\mu_1}(\mathcal{L}(\boldsymbol{\alpha}^t, \boldsymbol{\lambda}^t) - \mathcal{L}(\bar{\boldsymbol{\alpha}}^t, \boldsymbol{\lambda}^t)) \qquad (41)$$

*where*

$$\mu_1 := \frac{1}{\max\{16\theta_1 \Delta\mathcal{L}^0, 2\theta_1(1 + 4L_g^2)\}}.$$

*is the generalized strong convexity constant for function $\mathcal{L}(\boldsymbol{\alpha})$ with feasible domain $\mathcal{M}$, $\Delta\mathcal{L}^0$ is a bound on $\mathcal{L}(\boldsymbol{\alpha}^0) - \mathcal{L}(\bar{\boldsymbol{\alpha}}^0)$, $L_g$ is the local Lipschitz-continuous constant of $g(.)$ and $Q_{\max} = \max_{f \in \mathcal{F}} Q_f$.*

*Proof.* For each iteration $s$ of Algorithm 3, let $i$ be the chosen sample and suppose that out of $\nu$ partitions the one containing greedy factor satisfying (18) is chosen. We have

$$
\begin{aligned}
\mathcal{L}(\boldsymbol{\alpha}^{s+1}) - \mathcal{L}(\boldsymbol{\alpha}^s) &\leq \min_{\boldsymbol{\alpha}_{f^*}^s + \boldsymbol{d}_{f^*} \in \Delta^{|\mathcal{Y}_{f^*}|}} \langle \nabla_{\boldsymbol{\alpha}_{f^*}}\mathcal{L}, \boldsymbol{d}_{f^*} \rangle + \frac{Q_{\max}}{2}\|\boldsymbol{d}_{f^*}\|^2 \\
&= \min_{\boldsymbol{\alpha}_f^s + \boldsymbol{d}_f \in \Delta^{|\mathcal{Y}_f}} + \sum_{f \in \mathcal{F}_i} \langle \nabla_{\boldsymbol{\alpha}_f}\mathcal{L}, \boldsymbol{d}_f \rangle + \frac{Q_{\max}}{2}\left(\sum_{f \in \mathcal{F}_i} \|\boldsymbol{d}_f\|\right)^2
\end{aligned} \qquad (42)
$$

where the second equality follows from the optimality of $f^*$ w.r.t. (18). Then consider $i$ being uniformly sampled from $[n]$, and consider the probability that the partition containing greedy factor

$f^*$ is chosen, the expected descent amount is

$$E[\mathcal{L}(\boldsymbol{\alpha}^{s+1})] - \mathcal{L}(\boldsymbol{\alpha}^s)$$

$$\leq \frac{1}{n\nu} \left( \min_{\boldsymbol{\alpha}_f^s + \boldsymbol{d}_f \in \Delta^{|\mathcal{Y}_f}} \sum_{f \in \mathcal{F}} \langle \nabla_{\boldsymbol{\alpha}_f} \mathcal{L}, \boldsymbol{d}_f \rangle + \frac{Q_{\max}}{2} \sum_{i=1}^n \left( \sum_{f \in \mathcal{F}_i} \|\boldsymbol{d}_f\| \right)^2 \right)$$

$$\leq \frac{1}{n\nu} \left( \min_{\boldsymbol{\alpha}_f^s + \boldsymbol{d}_f \in \Delta^{|\mathcal{Y}_f}} \mathcal{L}(\boldsymbol{\alpha}^s + \boldsymbol{d}) - \mathcal{L}(\boldsymbol{\alpha}^s) + \frac{Q_{\max}}{2} \sum_{i=1}^n \left( \sum_{f \in \mathcal{F}_i} \|\boldsymbol{d}_f\| \right)^2 \right) \quad (43)$$

$$\leq \frac{1}{n\nu} \left( \min_{\beta \in [0,1]} \mathcal{L}(\boldsymbol{\alpha}^s + \beta(\bar{\boldsymbol{\alpha}}^s - \boldsymbol{\alpha}^s)) - \mathcal{L}(\boldsymbol{\alpha}^s) + \frac{Q_{\max}\beta^2}{2} \sum_{i=1}^n \left( \sum_{f \in \mathcal{F}_i} \|\bar{\boldsymbol{\alpha}}_f^s - \boldsymbol{\alpha}_f^s\| \right)^2 \right)$$

$$\leq \frac{1}{n\nu} \left( \min_{\beta \in [0,1]} \beta(\mathcal{L}(\bar{\boldsymbol{\alpha}}^s) - \mathcal{L}(\boldsymbol{\alpha}^s)) + \frac{Q_{\max}\beta^2}{2} \|\bar{\boldsymbol{\alpha}}^s - \boldsymbol{\alpha}^s\|_{2,1}^2 \right)$$

where $\bar{\boldsymbol{\alpha}}^s = \Pi_{\mathcal{S}}(\boldsymbol{\alpha}^s)$ is the projection of $\bar{\boldsymbol{\alpha}}^s$ to the optimal solution set $\mathcal{S}$. The second and last inequality is due to convexity, and the third inequality is due to a confinement of optimization domain. Then we discuss two cases in the following.

**Case 1:** $4L_g^2 \|B\boldsymbol{\alpha}^s - \boldsymbol{t}^*\|^2 < (\boldsymbol{b}^T \boldsymbol{\alpha}^s - s^*)^2$.

In this case, by the hoffman inequality (40), we have

$$\|\boldsymbol{\alpha}^s - \bar{\boldsymbol{\alpha}}^s\|_{2,1}^2 \leq \theta_1 (\|B\bar{\boldsymbol{\alpha}}^s - \boldsymbol{t}^*\|^2 + (\boldsymbol{b}^T \boldsymbol{\alpha}^s - s^*)^2)$$

$$\leq \theta_1 \left( \frac{1}{4L_g^2} + 1 \right) (\boldsymbol{b}^T \boldsymbol{\alpha}^s - s^*)^2 \quad (44)$$

$$\leq 2\theta_1 (\boldsymbol{b}^T \boldsymbol{\alpha}^s - s^*)^2,$$

since $\frac{1}{4L_g^2} \leq 1$. Then

$$|\boldsymbol{b}^T \boldsymbol{\alpha}^s - s^*| \geq 2L_g \|B\boldsymbol{\alpha}^s - \boldsymbol{t}^*\| \geq 2|g(B\boldsymbol{\alpha}^s) - g(\boldsymbol{t}^*)|$$

by the definition of Lipschitz constant $L_g$. Note that $\boldsymbol{b}^T \boldsymbol{\alpha}^s - s^*$ is non-negative since otherwise we have contradiction $\mathcal{L}(\boldsymbol{\alpha}^s) - \mathcal{L}^* = g(B\boldsymbol{\alpha}^s) - g(\boldsymbol{t}^*) + (\boldsymbol{b}^T \boldsymbol{\alpha}^s - s^*) \leq |g(B\boldsymbol{\alpha}^s) - g(\boldsymbol{t}^*)| - |\boldsymbol{b}^T \boldsymbol{\alpha}^s - s^*| \leq -\frac{1}{2}|\boldsymbol{b}^T \boldsymbol{\alpha}^s - s^*| < 0$. Therefore, we have

$$\mathcal{L}(\boldsymbol{\alpha}^s) - \mathcal{L}^* = g(B\boldsymbol{\alpha}^s) - g(\boldsymbol{t}^*) + (\boldsymbol{b}^T \boldsymbol{\alpha}^s - s^*)$$

$$\geq -|g(B\boldsymbol{\alpha}^s) - g(\boldsymbol{t}^*)| + (\boldsymbol{b}^T \boldsymbol{\alpha}^s - s^*) \quad (45)$$

$$\geq \frac{1}{2}(\boldsymbol{b}^T \boldsymbol{\alpha}^s - s^*).$$

Combining (43), (44) and (45), we have

$$\mathbb{E}[\mathcal{L}(\boldsymbol{\alpha}^{s+1})] - \mathcal{L}(\boldsymbol{\alpha}^s)$$

$$\leq \frac{1}{n\nu} \left( \min_{\beta \in [0,1]} -\frac{\beta}{2}(\boldsymbol{b}^T \boldsymbol{\alpha}^s - s^*) + \frac{2\theta_1 Q_{\max}\beta^2}{2}(\boldsymbol{b}^T \boldsymbol{\alpha}^s - s^*)^2 \right)$$

$$= \begin{cases} -1/(16\theta_1 Q_{\max} n\nu) & , 1/(4\theta_1 Q_{\max}(\boldsymbol{b}^T \boldsymbol{\alpha}^s - s^*)) \leq 1 \\ -\frac{1}{4n\nu}(\boldsymbol{b}^T \boldsymbol{\alpha}^s - s^*) & , o.w. \end{cases}$$

Furthermore, we have

$$-\frac{1}{16Q_{\max}\theta_1 n\nu} \leq -\frac{1}{16Q_{\max}\theta_1 n\nu(\mathcal{L}^0 - \mathcal{L}^*)}(\mathcal{L}(\boldsymbol{\alpha}^s) - \mathcal{L}^*) \quad (46)$$

where $\mathcal{L}^0 = \mathcal{L}(\boldsymbol{\alpha}^0)$, and

$$-\frac{1}{4n\nu}(\boldsymbol{b}^T \boldsymbol{\alpha}^s - s^*) \leq -\frac{1}{6n\nu}(\mathcal{L}(\boldsymbol{\alpha}^s) - \mathcal{L}^*) \quad (47)$$

since $\mathcal{L}(\boldsymbol{\alpha}^s) - \mathcal{L}^* \leq |g(B\boldsymbol{\alpha}^s) - g(\boldsymbol{t}^*)| + \boldsymbol{b}^T\boldsymbol{\alpha}^s - s^* \leq \frac{3}{2}(\boldsymbol{b}^T\boldsymbol{\alpha}^s - s^*)$. Since the bound (46) is much smaller than (47). For Case 1, we obtain

$$\mathbb{E}[\mathcal{L}(\boldsymbol{\alpha}^{s+1})] - \mathcal{L}^s \leq -\frac{\mu_1}{n\nu Q_{\max}}\left(\mathcal{L}(\boldsymbol{\alpha}^s) - \mathcal{L}^*\right) \tag{48}$$

where

$$\mu_1 = \frac{1}{16\theta(\mathcal{L}^0 - \mathcal{L}^*)}. \tag{49}$$

**Case 2:** $4L_g^2\|B\boldsymbol{\alpha}^s - \boldsymbol{t}^*\|^2 \geq (\boldsymbol{b}^T\boldsymbol{\alpha}^s - s^*)^2$.

In this case, we have

$$\|\boldsymbol{\alpha}^s - \bar{\boldsymbol{\alpha}}^s\|_{2,1}^2 \leq \theta_1\left(1 + 4L_g^2\right)\|B\boldsymbol{\alpha}^s - \boldsymbol{t}^*\|^2, \tag{50}$$

and by strong convexity of $g(.)$,

$$\mathcal{L}(\boldsymbol{\alpha}^s) - \mathcal{L}^* \geq \boldsymbol{b}^T(\boldsymbol{\alpha}^s - \boldsymbol{\alpha}^*) + \nabla g(\boldsymbol{t}^*)^T B(\boldsymbol{\alpha}^s - \bar{\boldsymbol{\alpha}}^s) + \frac{\rho}{2}\|B\boldsymbol{\alpha}^s - \boldsymbol{t}^*\|^2.$$

Now let $h(\boldsymbol{\alpha})$ be a function that takes value $0$ when $\boldsymbol{\alpha}$ is feasible and takes value $\infty$ otherwise. Adding inequality $0 = h(\boldsymbol{\alpha}^s) - h(\bar{\boldsymbol{\alpha}}^s) \geq \langle \boldsymbol{\sigma}^*, \boldsymbol{\alpha}^s - \bar{\boldsymbol{\alpha}}^s \rangle$ to the above gives

$$\mathcal{L}(\boldsymbol{\alpha}^s) - \mathcal{L}^* \geq \frac{\rho}{2}\|B\boldsymbol{\alpha}^s - \boldsymbol{t}^*\|^2 \tag{51}$$

because $\boldsymbol{\sigma}^* + \boldsymbol{b} + \nabla g(\boldsymbol{t}^*)^T B = \boldsymbol{\sigma}^* + \nabla\mathcal{L}(\bar{\boldsymbol{\alpha}}^s) = 0$ for some $\boldsymbol{\sigma}^* \in \partial h(\bar{\boldsymbol{\alpha}}^s)$. Combining (43), (50), and (51), we obtain

$$\begin{aligned}
&\mathbb{E}[\mathcal{L}(\boldsymbol{\alpha}^{s+1})] - \mathcal{L}(\boldsymbol{\alpha}^s) \\
&\leq \frac{1}{n\nu}\left(\min_{\beta\in[0,1]} -\beta(\mathcal{L}(\boldsymbol{\alpha}^s) - \mathcal{L}^*) + \frac{\theta_1(1 + 4L_g^2)Q_{\max}\beta^2}{\rho}\left(\mathcal{L}(\boldsymbol{\alpha}^s) - \mathcal{L}^*\right)\right) \\
&\leq -\frac{\rho}{n\nu\theta_1(1 + 4L_g^2)Q_{\max}}\left(\mathcal{L}(\boldsymbol{\alpha}^s) - \mathcal{L}^*\right)
\end{aligned} \tag{52}$$

Combining results of Case 1 (48) and Case 2 (52), and taking expectation on both sides w.r.t. the history, we have

$$E[\mathcal{L}(\boldsymbol{\alpha}^{s+1})] - \mathcal{L}(\boldsymbol{\alpha}^s) \leq -\frac{\mu_1}{Q_{\max}n\nu}\left(\mathcal{L}(\boldsymbol{\alpha}^s) - \mathcal{L}^*\right). \tag{53}$$

where

$$\mu_1 := \min\{\frac{1}{16\theta(\Delta\mathcal{L}^0)}, \frac{\rho}{\theta_1(1 + 4L_g^2)}\}.$$

Taking summation of (53) over iterates $s = 1...n$, we have

$$\begin{aligned}
E[\mathcal{L}(\boldsymbol{\alpha}^{t+1})] - \mathcal{L}(\boldsymbol{\alpha}^t) &\leq -\frac{\mu_1}{Q_{\max}n\nu}(\sum_{s=1}^{n}\mathcal{L}(\boldsymbol{\alpha}^s) - \mathcal{L}^*) \\
&\leq -\frac{\mu_1}{Q_{\max}\nu}(\mathcal{L}(\boldsymbol{\alpha}^{t+1}) - \mathcal{L}^*).
\end{aligned} \tag{54}$$

Rearranging terms gives the conclusion. $\qquad\square$

**Now we provide proof of Theorem 1 as follows.**

*Proof.* Let $\kappa = 4(1 + \nu)mQ/\mu_{\mathcal{M}}$. By lemma 3, 5, 4 and (30), we have

$$\begin{aligned}
&\Delta_d^t - \Delta_d^{t-1} + E[\Delta_p^t] - \Delta_p^{t-1} \\
&\leq \frac{-1}{1 + \kappa}\left(\mathcal{L}(\boldsymbol{\alpha}^t, \boldsymbol{\lambda}^t) - \mathcal{L}(\bar{\boldsymbol{\alpha}}^t, \boldsymbol{\lambda}^t)\right) + \frac{2\eta}{\rho}(\mathcal{L}(\boldsymbol{\alpha}^t, \boldsymbol{\lambda}^t) - \mathcal{L}(\bar{\boldsymbol{\alpha}}^t, \boldsymbol{\lambda}^t)) - \frac{\eta}{\tau}\Delta_d^t.
\end{aligned} \tag{55}$$

Then by choosing $\eta < \frac{\rho}{2(1+\kappa)}$, we have guaranteed descent on $\Delta_p + \Delta_d$ for each GDMM iteration. By choosing $\eta \leq \frac{\rho}{4(1+\kappa)}$, we have

$$(\Delta_d^t + E[\Delta_p^t]) - (\Delta_d^{t-1} + \Delta_p^{t-1})$$

$$\leq \frac{-1}{2(1+\kappa)} \left( \mathcal{L}(\boldsymbol{\alpha}^t, \boldsymbol{\lambda}^t) - \mathcal{L}(\bar{\boldsymbol{\alpha}}^t, \boldsymbol{\lambda}^t) \right) - \frac{\eta}{\tau} \Delta_d^t$$

$$\leq - \min \left( \frac{1}{2(1+\kappa)}, \frac{\eta}{\tau} \right) (\Delta_p^t + \Delta_d^t)$$

which then leads to the conclusion. $\qquad\square$

The proof for Theorem 2 follows the same line of above reasoning with step (55) replaced by application of Lemma 6 instead of Lemma 5.

# 7   Appendix C: Implementation details of FMO

## 7.1   C-1: Indicator Factor

Here we assume $\delta(y_j, \bar{y}_j)$ is constant for $\forall y_j \neq \bar{y}_j$ as in the case of Hamming error. Then we find maximizers of the 4 cases as following

(i) Visit $\boldsymbol{y}_f$ in descending order of $v(.)$ to find the first $\boldsymbol{y}_f$:$m_{if}(y_i) = 0$, $m_{jf}(y_j) = 0$.

(ii) $\forall y_j$:$m_{jf}(y_j) \neq 0$, visit $y_i$ in descending order of $v(.)$ to find the first $y_i$:$m_{if}(y_i) = 0$.

(iii) $\forall y_i$:$m_{if}(y_i) \neq 0$, visit $y_j$ in descending order of $v(.)$ to find the first $y_j$:$m_{jf}(y_j) = 0$.

(iv) Evaluate (26) for $\forall(y_i, y_j)$:$m_{if}(y_i) \neq 0$, $m_{jf}(y_j) \neq 0$.

Then $\boldsymbol{y}_f^*$ is returned as label ($\neq \bar{\boldsymbol{y}}_f$) of maximum gradient (26) among the 4 cases. One can verify the above procedure considers all labels that have potential to be $\boldsymbol{y}_f^*$. The complexities for (ii)-(iv) are bounded by $O(nnz(\boldsymbol{m}_{if})nnz(\boldsymbol{m}_{jf}))$, where $nnz(\boldsymbol{m}_{jf}) \leq |\hat{\mathcal{A}}_j^t|$. When BCFW adopts sampling without replacement, we have $|\hat{\mathcal{A}}_f^t| \leq t$. In practice, as $t$ keeps increasing, $|\hat{\mathcal{A}}_f^t|$ converges to a constant that depends on the optimal $nnz(\boldsymbol{\alpha}_f^*)$. Note $nnz(\boldsymbol{\alpha}_f^*)$ is equivalent to the number of labels $\boldsymbol{y}_f$ that attains the maximum of hinge loss (8), which is small in general as long as there are few labels with larger responses than the others.

Define $\mathcal{Y}_{NZ} = \{\boldsymbol{y}_f | m_{if}(y_i) \neq 0 \ \lor \ m_{jf}(y_j) \neq 0\}$ as the set of labels with messages from one of the variables involved, and $\mathcal{Y}_{Inc} = \{\boldsymbol{y}_f | \boldsymbol{y}_f \in \mathcal{Y}_{NZ} \land v(\boldsymbol{y}_f, \boldsymbol{x}_f) > v(\boldsymbol{y}_f', \boldsymbol{x}_f), \forall \boldsymbol{y}_f' \notin \mathcal{Y}_{NZ}\}$ as the subset being inconsistently ranked at the top in the multimap. The complexity of step (i) is $O(|\mathcal{Y}_{Inc}|)$, where

$$|\mathcal{Y}_{Inc}| \leq \max(|\mathcal{Y}_i||\hat{\mathcal{A}}_j^t|, |\mathcal{Y}_j||\hat{\mathcal{A}}_i^t|), \qquad (56)$$

which is sublinear to the size of factor domain $|\mathcal{Y}_f| = |\mathcal{Y}_i||\mathcal{Y}_j|$. Although the bound (56) is already sublinear to $|\mathcal{Y}_f|$, it is a *very loose bound*. In our experiments, we observed the average number of elements being visited at stage (i) is no more than 5 for problems of $|\mathcal{Y}_f|$ up to $10^7$, presumably because the inconsistency between factors is small in real applications.

## 7.2   C-2: Binary-Variable Interaction Factor

Similar to Appendix C-1, we're trying to find active factors with largest gradient. Here is the procedure.

(i) Visit $\boldsymbol{y}_f$ in descending order of $v(.)$ to find the first $\boldsymbol{y}_f$:$i \notin \mathcal{A}$ , $j \notin \mathcal{A}$.

(ii) $\forall j$:$j \notin \mathcal{A}$, visit $i$ in descending order of $v(.)$ to find the first $i$:$i \notin \mathcal{A}$.

(iii) $\forall i$:$i \notin \mathcal{A}$, visit $j$ in descending order of $v(.)$ to find the first $j$:$j \notin \mathcal{A}$.

(iv) Compute gradient for $\forall(i, j)$:$i \in \mathcal{A}$, $j \in \mathcal{A}$.

A similar reasoning as C-1 applies here for complexity analysis.