[Reviews · NeurIPS 2016]

Reviewer 1

Summary

The authors propose an improved method for training structural SVM, especially for problems with a large number of possible labelings at each node in the graph. The method is based on a dual factorwise decomposition solved with augmented Lagrangian, with the key speedup supported by a greedy factor search using special data structure. Experiments on several structured prediction tasks with large label set cardinality show very significant speedup over previous methods.

Qualitative Assessment

- Overall this is a well written paper. I particularly like how the authors target the problem of slow inference for factors with large domain by proposing a greedy search over the factors. The experiments also show that the speedup through solving this problem is significant. - The design of the algorithm is well-motivated and carefully thought out. For example, there are separate subroutines for dealing with factors with large domain and large number of factors. - How does the author set the parameter \rho (\eta) in the augmented Lagrangian? Is it adaptive or not? - What is the memory requirement for storing all the extra multimaps over factors? Since the cardinality of the label sets are large, the memory requirement might also be significant? - Have the authors considered caching the output y from previous iterations to avoid performing the expensive inference again? This could be helpful for the competing methods in the experiments.

Confidence in this Review

2-Confident (read it all; understood it all reasonably well)


Reviewer 2

Summary

This paper introduces an approach to learning of structural SVMs based on a direction method of multipliers. The starting point is to bring the primal form of the learning objective into a dual-decomposed representation (eq. 9), based on the LP relaxation of the inference problem. This representation is frequently used for inference in intractable graphical models, and has meanwhile also been exploited successfully for structured learning (e.g. in [13,14]). Basically, the problem is broken down into maximization of individual factors, as well as minimization with respect to the model parameters as well as Lagrange variables coupling the factors. Approaches then differ in how the coupling constraint are handled, and in how the minimization with respect to the model parameters is conducted. In the present work, the objective in (eq. 9) is again dualized, yielding an objective that is basically a sum of dual SVM objectives - which needs to optimized over variables subject to simplex constraints (as usual in the dual SVM representation), as well as additional coupling constraints resulting from the individual SVMs also being coupled in the primal representation. The coupling constraints are again handled by forming the Lagrangian; and updates with respect to the Lagrange multipliers are then performed by gradient descent. For minimizing the augmented Lagrangian function, the authors propose two different strategies: 1) Fully-corrective block-coordinate Frank-Wolfe (for factors/sub-SVMs of large cardinality) 2) Block-greedy coordinate descent (for a large number of factors/sub-SVMs) For factors of large cardinality, it is sometimes possible to find the maximizing configuration in sub-linear time. The authors give a few examples of where this is the case. However, these are not novel insights to my knowledge but have been exploited before. (See, e.g. Martins et al.: "AD3: Alternating Directions Dual Decomposition for MAP Inference in Graphical Models.", JMLR 16 (2015) 495-545). This maximization in sub-linear time can be exploited in the suggested learning algorithms. The authors perform an empirical comparison against several recent approaches, which suggest that their approach is highly competitive.

Qualitative Assessment

This manuscript definitely introduces some new ideas, and the empirical performance of the proposed algorithms seem strong. A lot of work was also put into the convergence proofs. These are non-trivial technical contributions that I find very valuable. However, I feel that the presentation lacks clarity and polish, and that there is a lack of rigor in the experimental evaluation: 1) There is no conclusion. It is not clear to me why the authors omitted it, as it seems there would have been some space for this still. The conclusion is very helpful for the reader in order to put things into perspective, so I strongly suggest to include one. 2) The description of the experiments is very brief and lacks important information. For instance, it is not clear to me how the \rho parameter of Soft-BCFW was chosen, or how the other competing algorithms were parameterized. I also do not see any mention of the step size parameter \gamma required by the authors' own algorithm. There needs to be a clean protocol for choosing hyper parameters. 3) I am not certain that the experimental comparison was fair. For the comparison in terms of computation time, the authors only plot their "GDMM-subFMO" approach, which exploits sub-linear maximization complexity of certain factor types. The competing algorithms, on the other hand, do not seem to exploit this, even though in my understanding, Soft-BCFW also only requires a factor maximization oracle, and hence could be made to benefit from exactly the same strategy. To better put things into perspective, I would like to see a comparison of "naive" GDMM (without subFMO) with the other algorithms, in terms of computation time. The claim of the authors is quite strong ("The proposed approach is orders-of-magnitude faster than the state-of-the-art training algorithms for Structural SVM"), so I think there must be a bullet-proof experimental validation that can also be reproduced by others. In the current state of the manuscript, I do not think that this is the case.

Confidence in this Review

3-Expert (read the paper in detail, know the area, quite certain of my opinion)


Reviewer 3

Summary

This paper presents a new algorithm for solving a structured learning formulation with decomposed output proposed in [13]. The proposed approach requires only O(1/\epslion) passes of factorwise oracle calls and perform well in practice.

Qualitative Assessment

[Technical] The paper provides nice theoretical and imperial analyses of the proposed algorithm. The algorithm is reasonable and well-motivated. [novelty] To my best knowledge, the O(1/\epsilon) convergence rate of optimizing Eq (9) is new and is impressive. [impact] Structured prediction approaches for large output domain Is useful. The proposed approach advances the training speed of the formulation proposed in [13]. [presentation] The paper is easy to follow and provide detailed explanations. It also provides several practical suggestions. - The empirical performance of the proposed approach is impressive. However, it is disappointed that the experiments only conducted on sequential tagging and multi-label tasks. The inference in the sequential tagging tasks can be easily solved by Viterbi (does methods like BCFW use the full oracle?) and there is no need to use approximate inference. Although multi-label tasks are complex, there are many other multi-label classification approaches. It is unclear the fully connected pairwise interaction is necessary and can achieve the best test performance. Nevertheless, the experiments are sufficient to support the claim of the paper. - I don't understand Eq (17) (indicator factor). What is the v function? Does this mean the score to the factor is a constant w.r.t w_f? - (minor) Please make the figures bigger.

Confidence in this Review

2-Confident (read it all; understood it all reasonably well)


Reviewer 4

Summary

The authors propose a risk minimization method for discrete factor graph models based on a Lagrangian decomposition into efficiently solvable per-factor problems. They provide theoretical results for their method and demonstrate speedups over Frank-Wolfe based methods in experiments.

Qualitative Assessment

Review summary. A difficult to read paper with a broadly applicable method; the experiments demonstrate large speedups over the state-of-the-art. Review. The presentation of the paper suffers from inadequate use of English which makes it a difficult read. The problem addressed is important and has significant prior work. An important prior work missing in the discussion is (Martins et al., "AD3: Alternating Directions Dual Decomposition for MAP Inference in Graphical Models", JMLR 2015). My understanding is that the basic decomposed problem (10)/(11) corresponds to (18) in the above JMLR paper but the current submission goes one step further an extend the setting to parameter estimation whereby now the subproblem quantities are linked to the parameters w. In (8) the definition of L_f happens implicitly and the authors should point out this out more clearly than is currently done in lines 95-96. The proposed methods are specialized to different types of factors (as in the JMLR paper, Section 5, and in work on Frank-Wolfe methods), and the result seems to be a practical method for parameter estimation in discrete factor graph problems. The theoretical results are interesting but lack a clear intuition: what is the "generalized geometric strong convexity constant" and can be it be controlled in an application? Is R_{\lambda} known apriori? A bit more intuition into these constants would be helpful in order to appreciate both Theorems. The experiments on NLP tasks demonstrate large improvements over the state-of-the-art Frank-Wolfe based learning methods. Minor details. Example corrections of English (there are too many to list): Line 1: "involve structured outputs". Line 2: "of a structured predictor". Line 3: "to an expensive inference oracle". Line 4: "by decomposing the training". Line 5: "replace an expensive structured oracle". Line 6: "with a Factorwise Maximization Oracle". etcetera.

Confidence in this Review

1-Less confident (might not have understood significant parts)


Reviewer 5

Summary

This paper presents a fast technique for learning structured SVM models for different settings of factors. The paper relaxes the structured SVM learning problem into an LP in a standard way (similar to Meshi et al and many others) breaking it into a summation of functions over individual factors connected via dual variables. The main contribution of the paper now follows. It proposes a greedy approach for maximizing w.r.t. a single factor (combinatorial problem) and choosing the direction of descent on the dual variables based on that. It now divides the problem space into two general categories that covers a large class of problems: 1) factors with large number of output configurations: it presents a technique that uses an active set of output variables for a given factor which is hopefully much smaller than all possible output configurations for a given factor. 2) large number of factors but each with a small number of output configurations: presents a technique that uses an active set of factors which will be smaller than all factors. For each of 1) and 2), the authors consider examples of commonly used factors and discuss how in practice gradient can be computed in time sublinear in the size of factor output configuration or the size of the number of factors.

Qualitative Assessment

This paper makes a solid contribution in structural SVM learning. The problems considered (large number of output configurations or large number of factors) are commonly faced in practice. The analysis is quite complete and useful. Some minor nits: 1) \lambda has been reused in equation 8 and 11 and I don't think they are the same variables. 2) Algorithms 1, 2, and 3 should all be accompanied by explanatory text. 3) The experiments can benefit from further discussion -- what were the sizes of active sets observed? Clear the authors are trading off time complexity for space complexity in the sublinear versions -- what were the observed memory burdens?

Confidence in this Review

3-Expert (read the paper in detail, know the area, quite certain of my opinion)


Reviewer 6

Summary

This paper considers the formulation of structured SVM via dual decomposition, and proposes a greedy direction method of multiplier to solve its dual problem. At each step, it calls a greedy block-coordinate descent algorithm to minimize the augmented Lagrangian function, and then it updates the Lagrangian multipliers to enforce consistency constraints. The paper describes two block-coordinate algorithms which either suit large domain size or large number of factors. They also implement the factorwise maximization oracle whose complexity can be sub-linear to the factor domain. Convergence analysis is also provided.

Qualitative Assessment

This paper is technically sound and potentially useful. According to the experiment results, the proposed algorithm is pretty efficient. However, the clarity of presentation can be improved. Detailed comments: - It's better to explicitly denote the input/output variables within each algorithm box. - Algorithm 3, step 2: why does "s" appear here?

Confidence in this Review

2-Confident (read it all; understood it all reasonably well)